

# PHYTOBASE: A global synthesis of open ocean phytoplankton occurrences

Damiano Righetti[1], Meike Vogt[1], Niklaus E. Zimmermann[2], Nicolas Gruber[1]

[1]Environmental Physics, Institute of Biogeochemistry and Pollutant Dynamics, ETH Zürich, Universitätstrasse 16, 8092 Zürich, Switzerland

[2]Dynamic Macroecology, Landscape Dynamics, Swiss Federal Research Institute WSL, 8903 Birmensdorf, Switzerland

*Correspondence to*: Damiano Righetti (damiano.righetti@env.ethz.ch)

**Abstract.** Marine phytoplankton are responsible for half of the global net primary production and perform multiple other ecological functions and services of the global ocean. These photosynthetic organisms comprise more than 4300 marine species, but their biogeographic patterns and the resulting species diversity are poorly known, mostly owing to severe data limitations. Here, we compile, synthesize, and harmonize marine phytoplankton occurrence records from the two largest biological occurrence archives (Ocean Biogeographic Information System; OBIS, and Global Biodiversity Information Facility; GBIF) and three recent data collections. The resulting PhytoBase data set contains over 1.36 million phytoplankton occurrence records (1.28 million at the level of species) for a total of 1711 species, spanning the principal groups of the *Bacillariophyceae*, *Dinoflagellata*, and *Haptophyta* as well as several other groups. This data compilation increases the amount of marine phytoplankton records available through the single largest contributing archive (OBIS) by 65%. Data span all ocean basins, latitudes and most seasons. Analyzing the oceanic inventory of sampled phytoplankton species richness at the broadest spatial scales possible, using a resampling procedure, we find that richness tends to saturate in the pantropics at ~93% of all species in our database, at ~64% in temperate waters, and at ~35% in the cold Northern Hemisphere, while the Southern Hemisphere remains underexplored. We provide metadata on the cruise, research institution, depth and date of collection for each record, and we include cell-counts for 195 339 records. We strongly recommend consideration of global spatiotemporal biases in sampling intensity and varying taxonomic sampling scopes between research cruises or institutions when analyzing the occurrence database. Including such information into statistical analysis tools, such as species distribution models may serve to project the diversity, niches, and distribution of species in the contemporary and future ocean, opening the door for a quantification of macroecological phytoplankton patterns. PhytoBase can be downloaded from PANGAEA, doi:10.1594/PANGAEA.904397 (Righetti et al., 2019a).

## 1 Introduction

Phytoplankton are photosynthetic members of the plankton, responsible for about half of the global net primary production (Field et al., 1998). While more than 4300 phytoplankton species have been described so far (Sournia et al., 1991), spanning at least six major clades (Falkowski, 2004), there are likely many more species living in the ocean, perhaps more than 10 000



(de Vargas et al., 2015). Some of these species (e.g. *Emiliania huxleyi*, *Gephyrocapsa oceanica*) are abundant and occur throughout the global ocean (Iglesias-Rodríguez et al., 2002), but a majority of marine plankton species form low-abundance populations (Ser-Giacomi et al., 2018) and remain essentially uncharted; i.e., the quantitative description of where they live, and where not, is rather poor. This biogeographic knowledge gap stems from a lack of a systematic global survey of

phytoplankton, as has been undertaken for inorganic carbon (WOCE/JGOFS/GOSHIP; Wallace 2001) or for trace metals (GEOTRACES; Mawji et al. 2015). Owing to logistic and financial challenges associated with internationally coordinated phytoplankton surveys, our knowledge of the biogeography of marine phytoplankton is, with a few exceptions (McQuatters-Gollop et al., 2015), mostly based on spatially very limited surveys or basin scale studies (e.g. Endo et al., 2018; Honjo and Okada, 1974). Global occurrence data on phytoplankton are unevenly distributed, incomplete in remote ocean areas, and orders

of magnitude higher in more easily accessed oceans, especially near coasts (Buitenhuis et al., 2013). Additional factors that have impeded progress in developing a good biogeographic understanding of the phytoplankton are difficulties in species identification, linked to their microscopic body size. This is well reflected in the current knowledge on the geographic distribution of phytoplankton species richness (Righetti et al., 2019b), which is much more limited compared to that of other marine taxa, such as zooplankton (e.g., Rutherford et al., 1999), fishes (e.g, Jones and Cheung, 2015), sharks (e.g., Worm et

al., 2005) or krill (e.g., Tittensor et al., 2010), even though many of these taxa also suffer from deficiencies in sampling efforts (Menegotto and Rangel, 2018).

Initial efforts to overcome the data sparseness and patchiness for phytoplankton by the MareDat project (Buitenhuis et al., 2012; Leblanc et al., 2012; Luo et al., 2012; O'Brien et al., 2013; Vogt et al., 2012) resulted in the compilation and synthesis of 118 phytoplankton species from 9738 sampling locations. While representing a large step forward, the coverage remained

relatively limited, largely owing to MareDat's focus on abundance data, motivated by the need to use the data for model evaluation and other quantitative assessments (Buitenhuis et al., 2013). But during these efforts, it became clear that there are at least an order of magnitude more data in archives around the world if one relaxed the abundance criterion and considered all observations that included presences. The potential for the use of presences to constrain e.g., phytoplankton community structure and richness, is large, as demonstrated by Righetti et al. (2019b), who recently produced the first global map of

phytoplankton species diversity. This application was also made possible thanks to the rapid developments in data mining and statistical analysis tools, such as species distribution models (SDMs) (Guisan and Zimmermann, 2000) that permit scientists to account for some of the limitations stemming from spatiotemporal sampling biases underlying species' occurrence data (Breiner et al., 2015; Phillips et al., 2009).

A key enabler for the compilation and synthesis of phytoplankton occurrences (presence or abundance records) is the existence

of two digital biological data archives, i.e., the Global Biodiversity Information Facility (GBIF; www.gbif.org), and the Ocean Biogeographic Information System (OBIS; www.obis.org). GBIF is the world's largest archive for species occurrence records, while OBIS is the largest occurrence database on marine taxa. Both archives have gathered a large number of phytoplankton occurrence records and make them freely available to the global community. In addition to MAREDAT (Buitenhuis et al.,



2013), marine surveys such as those conducted with the Continuous Plankton Recorder (CPR) (McQuatters-Gollop et al., 2015), the Atlantic Meridional Transect (AMT) (Aiken et al., 2000; Sal et al., 2013) and other programs provide relevant phytoplankton occurrence records, including data on species' abundance. A global synthesis of species occurrence records, including those from GBIF and OBIS has been attempted for upper trophic marine organisms, gathering 3.44 million records across nine taxa from zooplankton to sharks (Menegotto & Rangel 2018). But so far, no effort has been undertaken to bring the various sources together for the lowest trophic marine organisms, and merge them into a single harmonized database. This study aims to address this gap and to create PhytoBase, the world's largest open ocean phytoplankton occurrence database, which may substantially reduce the global limitations associated with phytoplankton undersampling.

The majority of the existing occurrence data of phytoplankton species have been collected via seawater samples of ~5–25 mL (Lund et al., 1958; Utermöhl, 1958), followed by microscopic specimen identification. Another key source of occurrence data is the continuous plankton recorder (CPR) program, in which plankton are sampled by filtering seawater onto a silk roll within a recorder device that is towed behind research– and commercial ships (Richardson et al., 2006). The plankton is then picked from the screens and identified by microscopy. DNA sequencing has become an alternative method to record and monitor marine phytoplankton at large scales (e.g. de Vargas *et al.* 2015; Sunagawa *et al.* 2015). However, within the recent global TARA Oceans cruise, ca. $\frac{1}{3}$ of DNA sequences of plankton from seawater could not yet be assigned to any taxon (de Vargas et al., 2015). For the most species-rich phytoplankton group (*Bacillariophyceae*), 58% of DNA sequences from seawater could be assigned to genus level in the same cruise (Malviya et al., 2016), but the majority of species have lacked reference DNA sequences needed for their identification. Additional factors have hampered the study of global phytoplankton biogeography: Some surveys lack resolution in terms of the species recorded (Richardson et al., 2006; Villar et al., 2015) and abundance information in terms of cells or biomass of species is often not available in the archived records (e.g. from GBIF). Second, the taxonomic identification and chronic undersampling of the species present in local communities via seawater samples (Cermeño et al., 2014) pose challenges, which can be resolved only by trained experts or larger sampling volumes. In addition, the rapidly evolving taxonomy (e.g. Jordan 2004) has led to varying use of nomenclature. These limitations need to be assessed and possibly overcome in a data synthesis effort.

Here, we compile 1 360 765 phytoplankton occurrence records (94.1% resolved to the level of species; *n* = 1716 species) and demonstrate that combining data from OBIS and GBIF increases the number of occurrence records by 52.7 % relative to the data solely obtained from OBIS. This gain increases to 65.2% when adding occurrence data from marine surveys, including MareDat (Buitenhuis et al., 2013), AMT cruises (Sal et al., 2013), and initial TARA Oceans results (Villar et al., 2015). With respect to species abundance information, we retain cell-count records whenever available from all sources, resulting in 195 339 quantitative entries. We harmonize and update the taxonomy between the sources, focusing on extant species and open ocean records. The resulting PhytoBase data set allows for studying global patterns in the biogeography, diversity, and composition of phytoplankton species. Using statistical SDMs, the data may serve as a starting point to examine species' niche



differences across all major phytoplankton taxa and their potentially shifting distributions under climate change. The data set can be accessed through PANGAEA, doi:10.1594/PANGAEA.904397 (Righetti et al., 2019a).

## 2 Compilation of occurrences

### 2.1 Data origin

To create PhytoBase, we compiled marine phytoplankton occurrences from five sources, including the two largest open access species-occurrence archives: the Global Biodiversity Information Facility (GBIF; www.gbif.org), and the Ocean Biogeographic Information System (OBIS; www.obis.org). These data were augmented with records from the Marine Ecosystem Data initiative (MareDat; Buitenhuis *et al.* 2013), with records from a marine micro-phytoplankton dataset (Sal et al., 2013), and with a subset of the data collected during the TARA Oceans cruise (Villar et al., 2015). We retrieved

phytoplankton records at the level "species" or below (e.g., "subspecies", "variety" and "form" were indicated by the taxon rank field in GBIF and OBIS downloads) for seven phyla or classes: *Cyanobacteria*, *Chlorophyta* (excluding macroalgae), *Cryptophyta*, *Myzozoa*, *Haptophyta*, *Ochrophyta*, and *Euglenozoa.* More specifically, among the *Ochrophyta*, we considered the classes *Bacillariophyceae*, *Chrysophyceae*, *Pelagophyceae,* and *Raphidophyceae.* Within the *Myzozoa*, we considered the class *Dinophyceae*. Within the *Euglenozoa*, we considered the class *Euglenoidea.* This selection of phyla or classes strived to

include all major marine phytoplankton taxa (following de Vargas et al., 2015 and Falkowski, 2004). In addition, we retrieved occurrences for *Prochlorococcus* and *Synechococcus* from all sources, as the latter two genera are often highly abundant (Flombaum et al., 2013), but rarely determined to the species level. Last, records from MareDat were considered for the functionally relevant genera *Phaeocystis*, *Richelia*, *Trichodesmium* and for non-specified picoeukaryotes. For simplicity, we refer to all genera as "species" in statistics presented herein.

For the taxa selected, occurrence data from GBIF and OBIS were first downloaded in December 2015 and updated in February 2017. Specifically, the initial retrieval of the GBIF data occurred on 7 December 2015 (using the taxonomic backbone from https://doi.org/10.15468/39omei, accessed on 14 July 2015), and the data were updated on 27 February 2017 (using an updated taxonomic backbone, accessed via http://rs.gbif.org/datasets/backbone, released 27 February 2017). The data from OBIS were first retrieved on 5 December 2015 (using the OBIS taxonomic backbone, accessed on 4 December 2015 via the R packages

*RPostgreSQL* and *devtools*) and updated for the selected taxa on 6 March 2017 (using the OBIS taxonomic backbone, accessed on 6 March 2017 via the R packages *RPostgreSQL* and *devtools*). The update in 2017 expanded the occurrences retrieved from GBIF substantially, with over 20 000 additional phytoplankton records stemming from an Australian CPR program alone (AusCPR, https://doi.org/10.1016/j.pocean.2005.09.011, accessed via gbif.org on 6 March 2017). We retained any GBIF sourced data that were retrieved in 2015, but deleted from GBIF before March 2017 (such as CPR data, with dataset key

83986ffa-f762-11e1-a439-00145eb45e9a).

In addition, we retrieved occurrences for the *Bacillariophyceae* and *Dinoflagellata* from initial TARA Oceans results (Villar et al., 2015; their Tables W8 and W9), we included the five phytoplankton papers from MareDat (Buitenhuis et al., 2012; Leblanc et al., 2012; Luo et al., 2012; O'Brien et al., 2013; Vogt et al., 2012) and the dataset of Sal et al. (2013). Additional

smaller datasets, as well as data processed by the TARA Oceans cruise or the Malaspina expedition (Duarte, 2015), may provide valuable additional data for a future synthesis, yet here we have focused on publicly available sources. These sources reflect decades to centuries of efforts spent on collecting global phytoplankton in situ data, until March 2017. A substantial amount of data from the CPR program (Richardson et al., 2006) are represented in the GBIF and OBIS archives and the data from Atlantic Meridional Transects (AMTs) 1 to 6 are represented in Sal et al. (2013), reflecting a substantial part of the data

from this monitoring program.

## 2.2 Data selection

We excluded occurrences from waters less than 200 m deep (Amante and Eakins, 2009), from enclosed seas (Baltic Sea, Black Sea or Caspian Sea), and from seas with a surface salinity below 20, using the globally gridded (spatial 1° x 1°) monthly climatological data of Zweng et al. (2013). This salinity-bathymetry threshold served to select data from open oceans,

excluding environmentally more complex and often more fertile near-shore waters.

### 2.2.1 Data accessed through GBIF and OBIS

We included GBIF data records on the basis of "human observation", "observation", "literature", "living specimen", "material sample", "machine observation", "observation" or "unknown", assuming that the latter was based on observation (see Table 1 for an overview of the metadata retained). With respect to OBIS data, we included data records on the basis of "O" or "D",

whereby "O" refers to observation and "D" to literature-based records. To filter out raw data of presumably inferior quality, records from OBIS and GBIF were removed: (i) if their year of collection indicated >2017 or <1800, (ii) if they had no indication on the year or month of collection (missing date) or (iii) if they had geographic coordinates outside the range -180 to 180 for longitude and/or outside -90 to 90 for latitude. However, as data from GBIF and OBIS were standardized to -180 to 180 degrees longitude (rather than 0 to 360 longitude East) and -90 to 90 degrees latitude, all records fulfilled the latter

criterion. Records with negative recording depths (<1% of data) were retained, assuming that the sign (usually positive) was mistaken.

### 2.2.2 Data accessed through MAREDAT

We included records at the species level for the *Bacillariophyceae* (Leblanc et al., 2012) and *Haptophyta* (O'Brien et al., 2013). In addition, we included all genus and species level records available for *Trichodesmium*, *Richelia* (Luo et al., 2012),

*Phaeocystis* (Vogt et al., 2012), *Synechococcus* (using the data-field "SynmL") and *Prochlorococcus* (using the data-field "PromL") (Buitenhuis et al., 2012). We included genus level records from the latter taxa, as they represent functionally important phytoplankton groups (Le Quéré, 2005), and as information on the presence and abundance of their cells, colonial

cells or trichomes often only existed at genus level (Buitenhuis et al., 2012; Luo et al., 2012; Vogt et al., 2012). In addition, we retained the records on picoeukaryotes, which were not determined to species or genus level (Buitenhuis et al., 2012). For

all taxa we retained the records with abundances (i.e., cell counts) reported larger than zero, while excluding records with zero entries or missing data entries, as our database focuses on presence-only or abundance records. In addition, we retained the species presence records on *Bacillariophyceae* host-cells from Luo et al. (2012). Given that data of the MareDat have been scrutinized previously, we flagged, rather than excluded reported years of data recording earlier than 1800 ($n = 564$; values 6, 10 or 11) and unrealistic day entries ($n = 58\,340$; values -9 or -1). The column "unrealisticDayOrYear" in the final PhytoBase

indicates such unrealistic day or year entries, originally associated with MareDat.

Harmonization of *Haptophyta* species names and taxonomy from MareDat (O'Brien et al., 2013) was guided by a synonymy table provided by O'Brien (*pers. comm.*) (Table A1). The harmonization of the *Bacillariophyceae* species names was in progress at the time of first data access (24 August 2015). The harmonization was completed and names corrected (Table A2). All data selected of MareDat were merged to a single dataset, containing the columns: "scientificName", "longitude",

"latitude", "year", "month", "day", "group", "Origin Database", "Cruise or station ID", "basis", "depth", and "rank".

### 2.2.3 Data accessed through Villar et al. (2015)

We compiled in situ presence records of species of *Bacillariophyceae* and *Dinoflagellata* from the tables W8 and W9 of Villar *et al.* (2015). These were the only records accessible at species level from the TARA Oceans cruise at the time of first data access (25 August 2015). We excluded species names containing "cf" (e.g *Bacteriastrum cf. delicatulum*), as such

nomenclature is typically used to refer to closely related species of an observed species. We retained all species ($n = 3$), which contained "group" in their names (e.g. *Pseudo-nitzschia delicatissima group*). *Tripos lineatus/pentagonus complex* was considered as *Tripos lineatus*. The cleaning of all spelling variants of original names from Villar et al. (2015) is presented in Table A3.

### 2.2.4 Data accessed through Sal et al. (2013)

The dataset of Sal et al. (2013) represents a highly complementary data source of phytoplankton occurrence records, i.e., it had no duplicated records with any of the other data sources considered. This data collection contains in situ samples subjected to a consistent methodology performed by the same taxonomist. We considered all records of the *Haptophyta*, *Bacillariophyceae*, *Dinophyceae*, *Peridinea*, *Dinophyceae* and *Dictyochophyceae* at species level or below (for the latter, we used the species name in the final database). These data included 5891 records, from 313 species and 541 samples.

### 2.3 Concatenation of source datasets

Column names or data-fields were adjusted and harmonized to establish compatibility in the dimensions of the different source datasets (Table 1). To retain relevant metadata, associated with specific source datasets, new columns containing these



**Table 1: Harmonization of original column names (data-fields) between data sources**

| Original column names | | | | | | | Final column names |
|---|---|---|---|---|---|---|---|
| GBIF (2015)* | GBIF (2017)* | OBIS (2015)** | OBIS (2017)** | MareDat | Villar et al | Sal et al | (all sources) |
| species | species | species | species | species | species | species | scientificName |
| basisOfRecord | basisOfRecord | basisofrecord | basisOfRecord | - | - | - | basisOfRecord |
| decimalLongitude | longitude | longitude | longitude | Longitude | Longitude | Lon | decimalLongitude |
| decimalLatitude | latitude | latitude | latitude | Latitude | Latitude | Lat | decimalLatitude |
| publishingOrgKey | - | - | - | - | - | - | publishingOrgKey_gbif§ |
| - | institutionCode | - | - | - | - | - | institutionCode_gbif§ |
| - | - | institutioncode | institutionCode | - | - | - | institutionCode_obis§ |
| - | - | - | - | Origin Database | - | - | originDatabase_maredat§ |
| datasetKey§§ | datasetKey§§ | - | - | - | - | - | datasetKey_gbif‖,§§,‖ SEP |
| - | - | collectioncode | collectionCode | - | - | - | collectionCode_obis‖§§ |
| - | - | - | resname | - | - | - | resname_obis‖ |
| - | - | resource_id§§ | resource_id§§ | - | - | - | resourceID_obis‖,§§ |
| - | - | - | - | CruiseorStationID | - | - | cruiseOrStationID_maredat‖ |
| - | - | - | - | - | - | Cruise | cruise_sal‖ |
| - | - | - | - | - | - | SampleID | sampleID_sal |
| taxonRank | taxonRank | - | - | rank | - | - | taxonRank‡ |
| taxonRank | taxonRank | - | - | cells l⁻¹, cells ml⁻¹# | cells ml⁻¹# | - | cellsPerLitre |
| - | individualCount¶ | - | observedindividualcount¶ | - | organismquantity | - | individualCount |
| year | year | yearcollected | year | Year | Date | Date | year |
| month | month | monthcollected | month | Month | Date | Date | month |
| day | day | daycollected | day | Day | Date | Date | day |
| depth | depth | depth | depth | Depth | Depth | Depth | depth |

* GBIF data were downloaded in 2015 (www.gbif.org; retrieved 7 December 2015) and 2017 (retrieved 27 February 2017)

** OBIS data were downloaded in 2015 (www.iobis.org; retrieved 5 December 2015) and 2017 (retrieved 6 March 2017)

‡ The "TaxonRank" field indicates the level of taxonomic resolution (species or genus) of observation records. Records of subspecies, varieties, and forms were generally retained in the data, but considered at the species level (using the genus and specific epithet). We obtained the species names for data from GBIF, OBIS, and Villar et al using the data-field "species".

§ These fields indicate the organization or institution by which original records were collected.

‖ These fields are indicators of different research cruises or resources, to which original records belonged.

# Values were transformed to cells per litre.

¶ The field "individualCount" and "observedindividualcount" had equivalent values for records that overlapped between GBIF and OBIS.

§§ datasetKey and resource_id are valuable to flag raw datasets related to "sediment cores" (or similar expressions) via API (OBIS, GBIF).

metadata were added to the source datasets. We then concatenated the different source datasets into a raw database which contained 1.51 million depth-referenced occurrence records of 3300 phytoplankton species (including five genera) and 247 385 sampling events (Table 2). Sampling events are thereby (and herein) defined as unique combinations of latitude, longitude, depth, and time (year, month, day) based on the highest available precision of occurrence records. We added the column

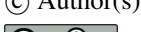



**Table 2: Summary statistics of the raw database by source**

| Source | Number of observations (%unique to source) | | Number of species[‖] (%unique to source) | | Number of observations (%unique to source) | | Number of species[‖] (%unique to source) | |
|---|---|---|---|---|---|---|---|---|
| | full data | | | | data with depth-reference | | | |
| GBIF | 970 927 | (65.6) | 3 977 | (60.4) | 908 995 | (64.2) | 2676 | (51.5) |
| OBIS | 853 981 | (60.5) | 2 305 | (25.2) | 823 968 | (60.1) | 1812 | (25.4) |
| MareDat | 102 621 | (94.6) | 123 | (1.1) | 102 467 | (94.7) | 123 | (1.5) |
| Villar *et al.* | 202 | (100.0) | 87 | (0.0) | 202 | (100.0) | 86 | (0.0) |
| Sal *et al.* | 5 891 | (100.0) | 314 | (0.0) | 5 867 | (100.0) | 313 | (0.1) |
| **Total** | **1 594 649** | | **4741** | | **1 511 351** | | **3300** | |

Numbers of observations (with % of observations unique to the source in parentheses) and the numbers of species (with % of species unique to the source in parentheses) are presented for each data source. Data of Picoeukaryotes (not identified to species or genus level) stemmed from MareDat and included 27 537 observations (all of which contained a depth-reference).

[‖] Species names are not harmonized with respect to synonyms or spelling variants.

"group" to the database, denoting to which phylum or class records belong: i.e., *Cyanobacteria*, *Bacillariophyceae*, *Chlorophyta*, *Chrysophyceae*, *Cryptophyta*, *Dinoflagellata*, *Euglenophyta*, *Haptophyta*, *Raphidophyceae* or picoeukaryotes, and the column "sourceArchive", indicating the source from which records were obtained (GBIF, OBIS, MAREDAT, VILLAR or SAL).

### 2.3.1 Extant species selection and taxonomic harmonization

We strived for a selection of occurrence data of extant phytoplankton species and a taxonomic harmonization of their multiple spelling variants (merging synonyms, while clearing misspellings or unaccepted names). This procedure included three cleaning steps:

(i) We discarded all species (and their data) that did not have any depth-referenced record. This choice was made on the basis of the argument that these species may have been predominantly recorded via fossil materials or have been associated with large uncertainty with respect to their sampling depth, which would infringe the scope of our database.

(ii) We extracted all scientific names (mostly at species level, including all synonyms and spelling variants) associated with at least one depth-referenced record from the raw database (Table 2). This resulted in 3300 names, which were validated against the taxonomic list of Algaebase (www.algaebase.org). Each name was verified by M. Guiry, the founder and director at Algaebase (M. Guiry, *pers. comm.*) in August 2017. The expert screening led to the exclusion of 459 names (and their data), which could not be traced back to any taxonomically accepted name at the time of query, and to the creation of a "synonymy table" in which each original name (including its potentially multiple synonyms and spelling errors) was matched to a corrected or accepted name.

(iii) We excluded fossil species (and their data), using information from Algaebase and the World Register of Marine Species (WoRMS; www.marinespecies.org, accessed August 2017) and we excluded species belonging to genera



with fossil types (www.algaebase.org) under the condition that these species lacked habitat information on both Algaebase and WoRMS. We assumed that the latter species have been collected based on sedimentary or fossilized materials. Species that were uniquely classified as "freshwater" on both Algaebase and WoRMS, were discarded, as these species are beyond the scope of our open ocean database. However, we retained the species classified as "freshwater", which had at least 24 open ocean (sect 2.2) records and thus were assumed to thrive also in marine
habitats: *Aulacoseira granulata, Chaetoceros wighamii, Diatoma rhombica, Dinobryon balticum, Gymnodinium wulffii, Tripos candelabrum, Tripos euarcuatus*. These cleaning steps led to a remaining set of 2041 original species names, synonyms or spelling variants, corresponding to 1716 taxonomically harmonized species (including 5 names of genera not resolved to the level of species).

### 2.3.2 Data merger and synthesis

We removed duplicate records, considering the columns "scientificName", "x", "y", "year", "month", "day", and "depth". Removing duplicates meant that any relevant meta-data of the duplicated (and hence removed) record were added to the meta-data of the record retained, either in an existing or additional column (e.g., information to which original dataset-keys the merged records belonged). We assigned the corrected and/or harmonized taxonomic species name to each original species name in the database on the basis of the synonymy table. We removed duplicates with respect to exact combinations of the
harmonized "scientificName", and "x", "y", "year", "month", "day", "depth". This resulted in the harmonized database containing 1 360 765 occurrence records (for which 95.8% had a depth-reference), 1716 species (including 5 genera not resolved to the level of species), and 242 207 sampling events (Table 3). We retained meta-information on the dataset ID, cruise number, and further attributes, when we removed duplicates with respect to harmonized names. In particular, we retained the original taxonomic names associated with each record in a separate column (taxonOriginal_"sourceArchive"), which
allows tracing back the harmonized name to its original name(s) and vice versa and will allow to implement future taxonomic name changes. Furthermore, we added the column "yearOfDataAccess", indicating the year of data download (2015, 2017 or both) and the column "containedWithinMLD_clim", which distinguishes records stemming from waters deeper than the oceanic mixed-layer (monthly climatology, de Boyer Montégut 2004) (11.5% of records) from those inside the mixed-layer. Besides the presence records, the final database includes 195 339 count records of individuals or cells, spanning 1127 species.
Among these, 335 species have counts with a volume reference ($n = 104\,327$ records), among which most of the counts stem from MareDat ($n = 94\,240$) and Sal *et al.* (2013) ($n = 5744$).

Last, we flagged sedimentary records, indicated by the added column "basisPresumablySedimentary". Although we excluded probably many records based on fossil materials during cleaning step (i), this does not exclude the possibility that occurrence records of extant species in the GBIF and OBIS source-datasets originated partially from sediment traps or sediment core
samples, rather than from seawater samples. Marine sediments can conserve phytoplankton shells that are exported to depth. We flagged phytoplankton records from OBIS and GBIF in the database associated with surface sediment traps or sediment





**Table 3: Summary statistics of the harmonized database by source**

| Source | Number of observations (%unique to source) | | Number of species (%unique to source) | | Number of observations (%unique to source) | | Number of species (%unique to source) | |
|---|---|---|---|---|---|---|---|---|
| | full data | | | | data with depth-reference | | | |
| GBIF | 790 224 | (54.9) | 1498 | (31.7) | 751 272 | (53.8) | 1447 | (31.3) |
| OBIS | 823 861 | (56.3) | 1325 | (21.7) | 796 924 | (56.0) | 1288 | (22.2) |
| MareDat | 101 969 | (94.7) | 123 | (2.6) | 101 816 | (94.8) | 121 | (2.7) |
| Villar *et al.* | 202 | (100.0) | 87 | (0.0) | 185 | (100.0) | 82 | (0.0) |
| Sal *et al.* | 5744 | (100.0) | 291 | (0.0) | 5721 | (100.0) | 282 | (0.0) |
| **Total** | **1 360 765** | | **1716** | | **1 303 783** | | **1716**[§] | |

Numbers of observations (with % of observations unique to the source in parentheses) and numbers of species (with % of species unique to the source in parentheses) presented for each data source.

[§] This number includes 1711 species and the genera *Phaeocystis*, *Trichodesmium*, *Richelia*, *Prochlorococcus* and *Synechococcus*. Data of Picoeukaryotes (which were not identified to species or genus level) were also retained, and stemmed from MareDat and included 27 537 observations, among which 10 725 records stemmed from the ocean mixed-layer.

cores by checking the metadata of each individual source dataset of GBIF (using the GBIF datasetKey) and OBIS (using the OBIS resourceID) sourced data, using the R package *rgibf* (using the function *datasets*) and the online portal of OBIS (http://iobis.org/explore/#/dataset, accessed 24 October 2018). This check resulted in the flagging of 2.7% of records. We did not attempt to clean or remove sediment-type records in the MareDat sources, assuming that information on sampling depth associated with the occurrence records of MareDat lead to thorough exclusion of sedimentary records previously. Data from Sal *et al.* (2013) and Villar *et al.* (2015) are based uniquely on seawater samples.

**3 Results**

**3.1 Data**

**3.1.1 Spatiotemporal coverage**

Phytoplankton occurrence records contained in PhytoBase cover all ocean basins, latitudes, longitudes and months (Fig. 1). However, data density is globally highly uneven (Fig 1B, C; histograms) with 44.7% of all records falling into the North
Atlantic alone, while only 1.4% of records originate from the South Atlantic, and large parts of the South Pacific basin are devoid of records (Fig. 1A). Analyzing the data by latitude (Fig. 1B) and longitude (Fig. 1C) reveals that sampling has been particularly thin at high latitudes (>70°N and S) during winter time. Occurrences cover a total of 18 863 monthly cells of 1° latitude × 1° longitude (using the World Geodetic System of 1984 as the reference coordinate system; WGS 84), which corresponds to 3.8% of all monthly ($n$ = 12 months) 1° cells of the open ocean (sect. 2.2). Without monthly distinction, records
cover 6098 spatial 1° cells, which is a fraction of 14.8% of all 1° cells of the open ocean.



**Figure 1: Global distribution of phytoplankton occurrence re cords of PhytoBase.** (**A**) Circles show the position of in situ occurrence

records (*n* = 1 360 765, including 1 280 257 records at the level of species), with the color indicating the source of the data. Map shading

indicates the extent of tropical (T >20°C; yellow), temperate (10°C≤ T≤ 20°C; snow-white), and cold (T <10°C; light-blue) seas, based on the annual mean sea surface temperature (Locarini et al. 2013). (**B-C**) Records plotted as a function of month and latitude (B) or longitude

(C). Colors of dots show the number of species detected in each "sample" (defined as any exact combination of time, location, and depth, in the final dataset). Histograms above panels (B-C) show the frequency of these samples by latitude (B), by longitude (C). (**D-E**) Histograms of sample frequency by year (D), by depth (E). Vertical yellow lines show the median.

Amounts of records are not evenly balanced between major phytoplankton taxa, and global sampling schemes differ between

these taxa (Fig. 2). CPR based observations are highly condensed in the North Atlantic (and to a lesser extent south of Australia) for the *Bacillariophyceae* and *Dinoflagellata* (Fig. 2A, B), but this aggregation is less clear for the *Haptophyta* (Fig. 2C), whose species have typically smaller cells compared to the former two groups. These three principal phytoplankton taxa have been well surveyed along the north-south AMT cruises, but they lack data in large areas of the South Pacific. Among the less species-rich taxonomic groups, including the *Cyanobacteria* (Fig. 2D) and *Chlorophyta*, global occurrence data coverage has

been sparser (Fig. 2D, E). Since all of the principal phytoplankton taxa (Fig. 2) are globally abundant and widespread, the phytoplankton occurrence patterns reported may closely reflect sampling efforts, and unlikely reflect a lack of phytoplankton.

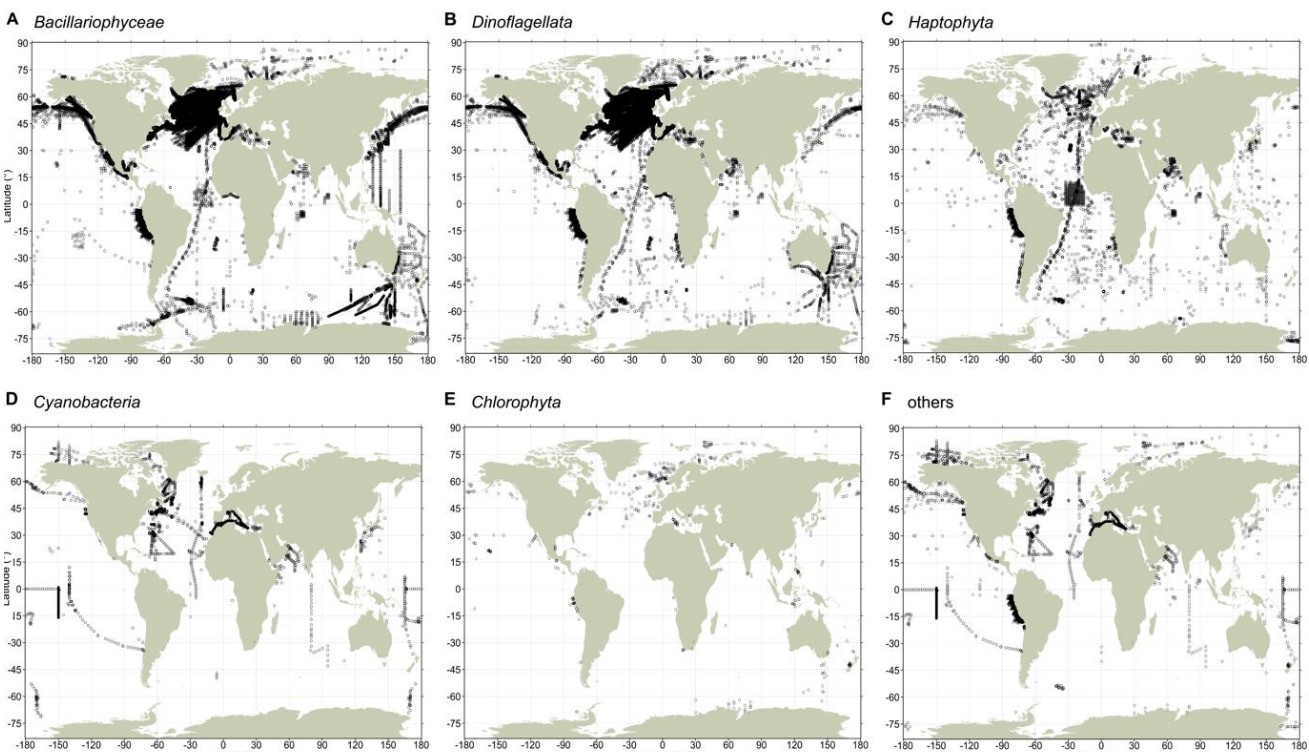

**Figure 2: Global distribution of phytoplankton occurrence records in PhytoBase for individual taxa.** Black circles show the distribution of in situ records for the five largest phyla or classes in the database that constitute 97.6% of all records (**A-E**) and for the

remaining taxa (**F**). Records may overlap at any particular location.



### 3.1.2 Environmental coverage

The phytoplankton occurrences compiled cover the entire temperature range and a broad part of nitrate and mixed layer conditions found in the global ocean (Fig. 3A, B). To visualize such environmental data coverage, figure 3 matches the

**Figure 3: Phytoplankton records in environmental parameter space.** (**A-B**) Dots display in situ records ($n = 1\ 360\ 765$) as a function of sea temperature and nitrate concentration (A), and as a function of mixed-layer depth (MLD) and nitrate concentration (B). The scale is logarithmic for MLD and nitrate. Shading indicates the relative frequency of environmental conditions appearing in the ocean, with darker grey shade indicating higher frequency. The colors of the dots denote the source of data, indicating complementarity or overlap of environmental sampling space between archives (**C-D**) Show the subset of records that contain information on species' cell counts with a valid volume basis ($n = 104\ 327$), stemming largely from MareDat.





occurrence records in PhytoBase with climatological sea surface data on nitrate (Garcia *et al.* 2013), temperature (Locarini *et al.* 2013), and mixed-layer depth (de Boyer Montégut, 2004) at monthly 1° × 1° resolution. Records are concentrated in areas with intermediate conditions, which are relatively more frequent at the global scale (gray shade; Fig. 3A, B). Data including cell-counts (13.1% of all records) show a similar coverage as the full set (Fig. 3A, B), but data are much thinner (Fig. 3C, D).

### 3.1.3 Taxonomic coverage

We assessed what fraction of the known marine phytoplankton species (Falkowski, 2004; Jordan, 2004; de Vargas et al., 2015) is represented by PhytoBase. The records compiled include all major taxa of marine phytoplankton known ($n$ = 9 phyla or classes), including the *Bacillariophyceae*, *Dinoflagellata*, and *Haptophyta*. Records span roughly half of the known marine species of the *Haptophyta* (Jordan, 2004) and a similar fraction of the known marine *Bacillariophyceae* and *Dinoflagellata* species (Table 4). By contrast, species of the less species-rich taxa tend to be more strongly underrepresented and account for a relatively small fraction (~7-10%) of all species in PhytoBase.

Record quantities in PhytoBase are unevenly distributed between individual species (Fig. 4). Half of the species in PhytoBase contain at least 29 presence records, but multiple species contribute one or two records each (Fig. 4A). The species with less than 29 records account for as little as 0.53% of all species records in PhytoBase. Similarly, half of all genera contain at least 107 records each, while genera with less than 107 records each contribute as little as 0.34% to the total of records. A similar data distribution applies to the subset of species ($n$ = 335), for which cell-count records (with volume reference) are available (Fig. 4B). Half of these species contribute at least 16 records, and half of all genera ($n$ = 127) contribute at least 73 records.

### 3.1.4 Completeness of species richness inventories at large spatial scales

We analyzed the ocean inventory of phytoplankton species richness in the database for three different regimes of ocean temperature by means of species accumulation curves (SACs) (Thompson and Withers, 2003) (Fig. 5). These curves present the cumulative species richness detected as a function of sampling effort (or survey area) and are expected to increase asymptotically before they saturate above a certain threshold of sampling effort (i.e., when the system has been exhaustively sampled). Using the number of sampling events (i.e., unique combinations of time, depth, location in our database, x-axis) as a surrogate for sampling effort, we find that the richness detected (y-axis) and the completeness of species richness detection (degree of saturation) differ notably between regimes. In the Southern temperate (Fig. 5E) and cold (Fig. 5F) ocean, richness has been strongly incompletely sampled with respect to total species (black lines) or key taxa (colored lines). By contrast, SACs in the Northern Hemisphere start to saturate at ~40 000 samples, suggesting that sampling efforts have recorded a majority of the species. Specifically, the SACs suggest that species richness will saturate at around ~1500 species in the tropical regime (>20°C), at ~1100 species in northern mid latitudes (≥10°C, ≤ 20°C), and at ~600 species in the cold Northern Hemisphere (>10°C). Compared to the ~1700 species considered in our database, this represents 93%, 64% and 35% of all



**Table 4: Statistics on data collected and species contained in the database for key taxa**

| Taxon | Range (mean) of known marine species number | Sources contributing to database | Number of records in database | Number of species or taxa (% of total species in database) | % of known marine species number | |
|---|---|---|---|---|---|---|
| *Bacillariophyceae* | 1800[†]-5000[§] (3400) | GBIF, OBIS, MareDat, Villar et al., Sal et al. | 699 111 | 705 (41.1) | 14-39 | 350 |
| *Dinoflagellata* | 1780[†]-1800[§] (1790) | GBIF, OBIS, Villar et al., Sal et al. | 527 293 | 778 (45.3) | 43-44 | |
| *Haptophyta* | 300[†,‖]-480[§] (360) | GBIF, OBIS, Sal et al., MareDat | 47 183 | 166 (9.7) | 34-55 | |
| *Chlorophyta* | 100[§]-128[†] (114) | GBIF, OBIS | 1448 | 30 (1.7) | 20-25 | 355 |
| *Chrysophyceae* | 130[†]-800[§] (465) | GBIF, OBIS, Sal et al. | 2111 | 13 (0.8) | 1-8 | |
| *Cryptophyta* | 78[†]-100[§] (89) | GBIF, OBIS | 2312 | 11 (0.6) | 4-5 | |
| *Cyanobacteria* | 150[§] | GBIF, OBIS, MareDat | 50 273 | 6 (0.3) | 3 | |
| *Euglenoidea* | 30[§]-36[†] (33) | GBIF, OBIS | 701 | 3 (<0.2) | 6 | |
| *Raphidophyceae* | 4[†]-10[§] (7) | GBIF, OBIS | 9 | 4 (0.2) | 20-50 | 360 |
| Picoeukaryotes | No reference | MareDat | 27 537 | 1 | - | |
| **Total** | **4530[†,¶]-16 940[§] (10 735)** | **5** | **1 360 765** | **1717** | **10-38** | |

The table summarizes the occurrence records for the ten major taxa in the database and describes to what degree the species in each taxon represent the total number of marine species known (for which exact numbers are still debated; we therefore provide upper and lower bounds, and mean values in parentheses).

§ Falkowski et al. (2004). Estimate includes coastal taxa and open ocean taxa, while this paper focuses on occurrence data collected from the open oceans.

† de Vargas et al. (2015)

‖ Jordan et al. (2004)

¶ Estimate excludes prokaryotes (De Vargas et al. 2015). A number of 150 prokaryotes (Falkowski et al. 2004) was added to obtain the mean total species number.

species, respectively. However these estimates only represent the fraction of species detectable via light microscopy, and other methods underlying our database, preferentially omitting very rare or small species (Cermeño et al., 2014; Ser-Giacomi et al., 2018; Sogin et al., 2006). Thus, the richness will likely increase (at low rates) with additional sampling efforts. Theoretical models have suggested that communities with many rare species lead to SACs with "low shoulders" meaning that SACs have a long upward slope to the asymptote (Thompson and Withers, 2003), consistent with our SACs (Fig. 5).

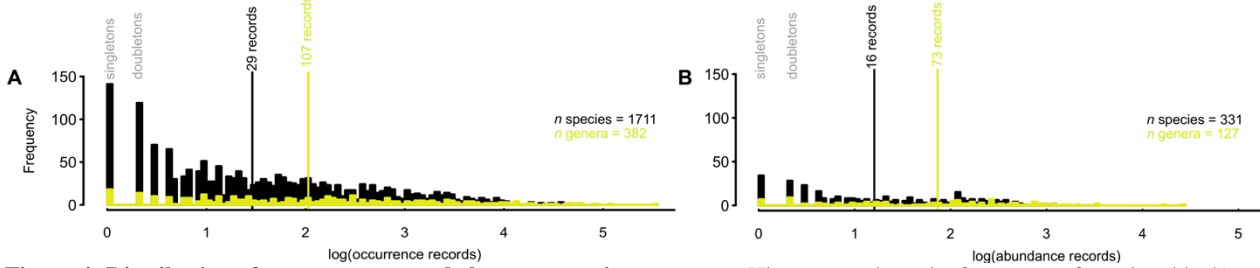

**Figure 4: Distribution of occurrence records between species or genera.** Histograms show the frequency of species (black) and genera (yellow) with a certain amount of presence (**A**) or abundance records (**B**) separately. Vertical lines (black, yellow) indicate the median value. *X*-axes are logarithmic to the base ten.



**Figure 5: Accumulation of species richness as a function of sampling effort, by region.** Curves show the cumulative species richness as a function of samples (i.e., unique combinations of space, time and depth in the database, drawn at random) drawn at random from the database, using 100 runs (shadings around the curves indicate ± 1 S.D). Shown are species accumulation curves for all species (black) and three major taxa (colours) for the tropics (T >20°C) (**A**), temperate seas (10°C≤ T≤ 20°C) of Northern Hemisphere (**B**), cold seas (T< 10°C) of Northern Hemisphere (**C**), temperate seas (10°C≤ T≤ 20°C) of Southern Hemisphere (**D**), cold seas (T< 10°C) of Southern Hemisphere (**E**), see background shade in map of figure 1.

### 3.1.5 Species richness documented within 1° cells

To explore how completely species richness has been sampled at much smaller spatial scales, we binned data at 1° × 1° resolution, and analyzed the number of species in the pooled data per cell as a function of sampling effort. Hotspots in directly observed phytoplankton richness at the 1° cell level emerge in near-shore waters of Peru, around California, south-east of Australia, in the North Atlantic, along AMT cruises, and along research transects south of Japan (Fig. 6A). The species richness detected per 1° cell is positively correlated with sampling effort, using the number of samples collected per cell as a surrogate of sampling effort (Spearman's $\rho = 0.47$, $P < 0.001$). In particular, richness of *Bacillariophyceae* ($\rho = 0.88$, $P < 0.001$) and of *Dinoflagellata* ($\rho = 0.92$, $P < 0.001$), is positively correlated with effort, while this is less so for *Haptophyta* ($\rho = 0.27$; $P < 0.001$). Analyzing species richness as a function of "sampling events" for different thermal regimes separately reveals that



**Figure 6: Species richness observed within 1° cells.** (A) Global map visualizing the species richness detected within each 1° latitude x 1°
longitude cell of the ocean. (The means of four 1° cells are depicted at 2°-resolution). (**B-E**) The number of species detected within each 1°-
cell is plotted as a function of sampling effort per cell (i.e., number of sampling events, defined as unique combinations of position, time and
depth in the database), with colours indicating data originating from different regions: tropical (T >20°C; yellow), temperate (10°C≤ T≤
20°C; snow-white), and polar 1° cells (T< 10°C; light-blue), as defined by the annual mean temperature at sea surface (Locarini *et al.*, 2013;
see shading of map in Fig. 1). The richness-effort relationship is shown for all taxa (B), and major taxa separately (C-E).

tropical areas (yellow dots; Fig. 6B-E) yield higher cumulative per-cell richness at moderate to high sampling effort (more
than ~50 samples), than temperate (grey dots) and polar areas (blue dots) (Fig. 6B-E). Although data are thin and scattered,



species richness in cold areas tends to saturate at ~70 species per cell (Fig. 6B; blue dots) at an effort of ~500 samples collected per cell. In contrast, species richness of the tropical areas tends to reach ~290 species per cell at the same effort (~500 samples). This suggests that tropical phytoplankton species richness at the cell level is about 4 times higher than that of the cold northern regime, but richness may further increase with additional sampling effort. Analyzing the data of the major taxa separately suggests that ~200 species of *Bacillariophyceae* and *Dinoflagellata* can be collected at high sampling effort (~500 samples), yet data are very sparse for the *Haptophyta*, which generally lack 1° cells with more than 100 samples collected (Fig. 6E).

The analysis of detected species richness per 1° cells suggests that roughly $^1/_3$ to $^1/_5$ of all species inventoried in the tropical or polar regime through our database (Fig. 5) can be detected within a single 1°-cell of these regimes at high sampling effort (~500 samples). This result is in coarse agreement with the result obtained at the large spatial scale (Fig. 2.5 A-C), showing that cumulative detected richness in the tropical regime is close to 3 times the richness detected in the (northern) cold regime.

### 3.1.6 Comparative spatial and taxonomic analysis of source datasets

We analyzed the sources obtained from within the GBIF archive as an exemplary case for a more detailed examination of original source data coverage, as GBIF provides relatively detailed information on its sources via dataset keys. The single largest contributing source dataset to GBIF data obtained is CPR, which covers the North Atlantic and North Pacific (Fig. 7A-D; brown dots), and parts of the ocean south of Australia (Fig. 7A-D; blue dots). CPR records obtained via GBIF contribute 33.8% to all records in PhytoBase. CPR data show relatively low species numbers captured on average per "sample" (Fig. 7I), with samples being defined as exact combinations of position, depth, and time in the data. This may be owing to the continuous collection of species or incomplete reporting of taxa. The mesh size of the silk employed in CPR (270 μm) under-samples small phytoplankton species (<10 μm). Yet, small species nevertheless get regularly captured in CPR, as they get attached to the screens (Richardson et al., 2006). Within the 16 largest source datasets obtained via GBIF, the average number of species collected per "sample" is below four for the CPR program and increases to >40 for other source datasets (Fig. 7I). These 16 datasets (excluding datasets with sedimentary records) presented in figure 7 demonstrate how strongly the taxonomic resolution differs between samples of individual surveys or cruises. By latitude, different surveys or cruise programs thus contribute to the occurrences in PhytoBase to a varying degree (Fig. 7E-H). Systematic differences in the species detected per sample and the varying contribution of sources to the database along latitude (Fig. 7E-H) are important considerations when, for example, analyzing species richness directly.

Analyzing the 16 largest source data sets from GBIF (Fig. 7) in environmental parameter space reveals that different regimes of global sea surface temperatures, nitrate levels, and mixed-layer depths have been sampled (Fig. 8). GBIF data sets collected in the tropics and subtropics (mean temperature of sampling of 20° C or higher; Fig. 8A) tend to be associated with higher taxonomic detail (~25 species detected per sample on average; Fig. 7I), compared to datasets collected in colder areas. Yet, this likely also reflects an overall higher number of species occurring in tropical areas (Figs. 5A) than in extratropical ones.

**Figure 7: Spatial extent of the 16 largest datasets from GBIF and average per-sample richness.** (**A-D**) Maps display the spatial distribution of the 16 biggest contributing datasets to the GBIF-sourced data in the database, showing each season separately. The datasets presented comprise 54.8 % of all records and 94.0 % of GBIF-sourced records. GBIF data is shown as an exemplary case, as it contributes a variety of source datasets defined by dataset keys (DatasetKey_gbif). Panels (**E-H**) show the importance of contributing datasets, by latitude. The width of coloured sub-bars reflects the amount of occurrences from each dataset, in 5°-latitude bands. Panels (E–H) correspond to data shown in (A–D). (**I**) Box plots show the mean (thick vertical lines) species richness detected in samples of each dataset. Boxes show the first and third quartiles for richness distribution around the mean. Whiskers show 2.5 times the inter-quartile range. Note that the same analysis may be performed for OBIS-sourced data using the field "ResourceID_obis" in the database.





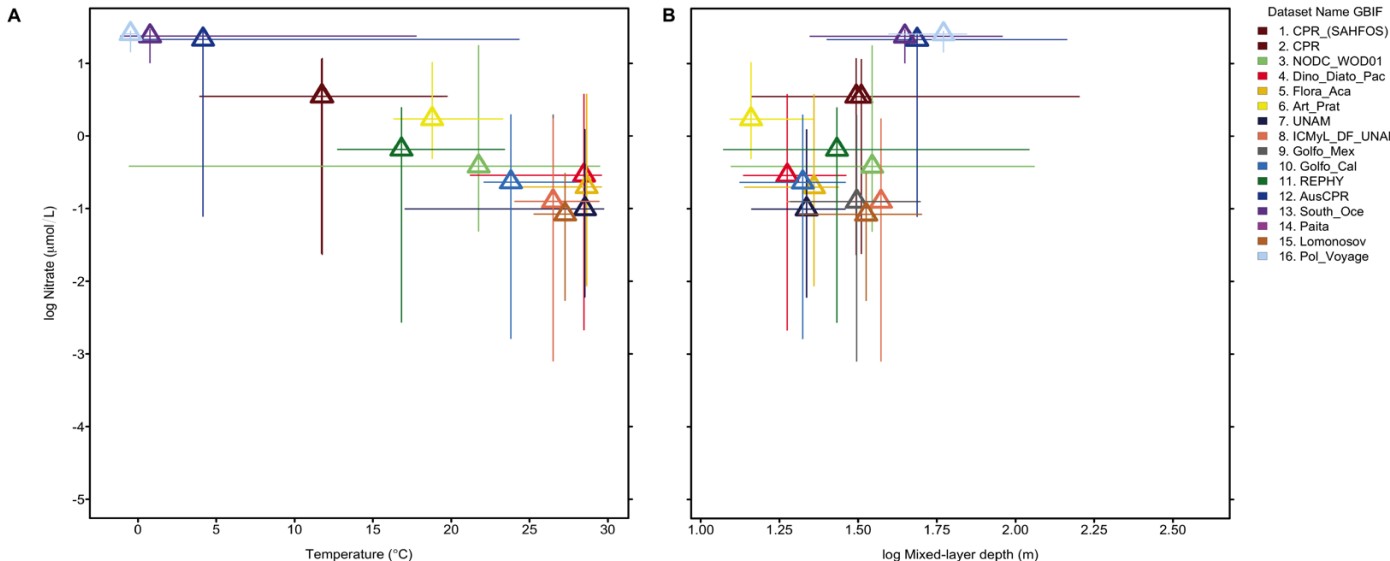

**Figure 8: Environmental range of the 16 largest datasets from GBIF.** (A-B) The range of 16 datasets contained within GBIF-sourced data, and the range of the dataset from Sal et al. (2015), are represented by thin lines in parameter space: (A) temperature *vs.* logarithmic

nitrate concentration in the surface ocean, and (B) logarithmic mixed-layer depth vs. logarithmic nitrate (using climatological environmental data from Garcia *et al.* 2013; Locarini *et al.* 2013; de Boyer Montegut, 2004; matching with records at monthly climatological 1°-resolution). Lines span the minimum to maximum environmental condition associated with records in each dataset, and triangles display the mean environmental condition of all records per dataset.

### 3.1.7 Sensitivity of data to taxonomic harmonization and coordinate rounding

While GBIF-derived data contributed more records (970 927) than OBIS (853 981) in the raw data (Table 2), this relative contribution changed after taxonomically harmonizing the database. GBIF finally contributed 790 224, while OBIS contributed 823 861 records to the harmonized PhytoBase. This shows that the exclusion of non-marine, fossil or doubtful species and the taxonomic harmonization, were more stringent for GBIF-sourced than OBIS-sourced data.

We tested to what degree the number of unique records in the final and harmonized database changed when rounding decimal

positions in the raw data in each of the sources prior to their merger. We find that the total number of unique records in PhytoBase declines continuously from 1.36 million to 1.07 million, when rounding the coordinates of records in the raw data to the 6th, 5th, 4th, 3rd, and 2nd decimal place. This result may be explained by the fact that large parts of the data come from CPR. The records of CPR start to be binned into coarser "samples" when rounding their decimal positions. The harmonized database (without coordinate rounding) gained 65.2% occurrence data, relative to its largest source archive. This gain was

similar in the non-harmonized database and ca. 73% when rounding coordinates to varying decimals. This shows that different sources contribute complementary records to PhytoBase, regardless of coordinate rounding to varying decimals.



## 4 Discussion

### 4.1 Data coverage, uncertainties, and recommendations

Spatiotemporal data on species occurrence are an essential basis to determine, assess, and forecast species' distributions and
to understand the drivers behind these patterns. Following recent calls to gather species occurrences into global databases
(Edwards, 2000; Meyer et al., 2015), we merged observational data of marine phytoplankton from three marine data sources
and from the two largest open-access biological data archives into PhytoBase. This new database contains 1 360 765 records
(1 280 257 records at the level of species) describing 1716 species across nine major taxonomic groups. Our effort addresses a
gap in analyzing marine species occurrence data at the global scale, as previous studies of marine taxa (Tittensor *et al.* 2010;
Chaudhary *et al.* 2016; Menegotto & Rangel 2018) had no easy access to data sufficiently complete for global analyses of
phytoplankton. The synthesis and harmonization of GBIF data with OBIS and other sources results in a substantial gain of
phytoplankton occurrence records (> 60% additional records), relative to phytoplankton records residing in either of the two
archives. The harmonization of data from different sources therefore substantially expands the empirical basis of phytoplankton
records from open access data archives.

PhytoBase presents, to the best of our knowledge, the currently largest global database of marine phytoplankton species
occurrences. However, two main limitations remain: First, the global data density is spatially highly uneven and important
gaps persist across large swaths of the ocean, e.g., in the South Pacific and the central Indian. Second, the sampling efforts
across larger taxa or species, and across different size classes differ widely. This is a result of the large differences in sampling
methods, sampling volumes, and taxonomic expertise (Cermeño et al., 2014). Results show that the average number of species
detected per sample varies from three to above 40 between different cruises or programs. A global spatial bias in collection
density of marine species has been similarly found for heterotrophic taxa (Woolley *et al.* 2016; Menegotto & Rangel 2018),
but sampling biases and divergent sampling protocols between cruises may be even more common for phytoplankton.

Owing to these limitations, we recommend that direct analyses of the database be undertaken and interpreted with caution. For
example, our data analysis has shown that direct species richness estimates are sensitive to the number of sampling events. In
addition, many species have very low numbers of occurrences in the database, making any inference about their ecological
niche or their geographic distribution very uncertain. Thus, without careful screening and checking of the data, the
characterization of biogeographies at the species level might be highly biased.

Statistical techniques such as rarefaction (Rodríguez-Ramos et al., 2015), randomized resampling (Chaudhary et al., 2017),
analysis of sampling gaps (Woolley *et al.* 2016; Menegotto & Rangel 2018), and species distribution modeling (Zimmermann
and Guisan, 2000) may be implemented to overcome these limitations. The latter statistical technique may be particularly
promising, as species distribution models can be set up to account for variation in presence data sampling bias (Phillips et al.,
2009) and data scarceness (Breiner et al., 2015). Based on observed associations between species' occurrences and
environmental factors (Guisan and Thuiller, 2005), these models estimate the species' ecological niche, which is projected into

Earth System
Science
Data

geographic space, assuming that the species' niche and its geographic habitat are directly interrelated (Colwell and Rangel,
2009). Another advantage of species distribution models is that they can circumvent geographic sampling gaps through a
niche-for-space substitution (a spatial projection of the niche), as long as environmental conditions relevant to describe the
niche of species have been sufficiently well sampled and the species fills its ecological niche. This is the approach used by
Righetti et al. (2019b), building on a large fraction of the PhytoBase (77.6% of the records, falling into the mixed-layer), to
analyze global richness patterns in phytoplankton.

Sampling efforts based on DNA sequencing have become an alternative approach to characterize phytoplankton biogeography
(de Vargas et al., 2015). These data have two advantages over the traditional taxonomic sampling data presented: First, the
sensitivity of metagenomic methods to detect rare taxa is much higher compared to traditional sampling. Second, metagenomic
data have been collected in a methodologically consistent way in recent global surveys, such as TARA Oceans (de Vargas et
al., 2015). But there are also drawbacks associated with DNA based methods. A large (current) disadvantage of current
metagenomic data is the lack of catalogued reference gene-sequences for most species. As a result, the majority of the
metagenomics sequences can only be determined to the level of genus (Malviya et al., 2016). However, we expect that an
integration of detailed genetic data with traditional sampling data may soon become possible, pushing phytoplankton
occurrences availability and taxonomy forward massively. At any point in the future, changing phytoplankton taxonomic
nomenclature can be easily considered and implemented in PhytoBase, as we retain the original name variants or synonyms
from the raw data sources together with the harmonized name variants for each record in PhytoBase.

## 4.2 Data use

Our data compilation and synthesis product PhytoBase was designed to support primarily the analysis of the distribution,
diversity, and abundance of phytoplankton species and related biotic or abiotic drivers in macroecological studies. But
PhytoBase is far from limited to this set of applications, and may include the analysis of ecological niche differences between
species or clades, linkages between species' ecological niches and phylogenetic or functional relatedness, current or future
spatial projections of species' niches, tests on whether presence-absence patterns of multiple species can predict community
trait-indices, studies on how well species' traits predict spatial patterns of species, or joint analyses of species' distribution and
trait data to project trait biogeographies. The database may also be used to validate the increasingly complex marine ecosystem
models included in regional to global climate models.

The accuracy of data analyses may be limited by sampling biases underlying PhytoBase, including the spatiotemporal variation
in sampling efforts and varying taxonomic detail between data sources or research cruises. The latter limitation might be
alleviated by considering different methodologies associated with varying cruises or collecting organisations in spatial
analyses. Where possible, we thus retained the information on the original dataset ID or the dataset key along with each
occurrence record in the database. Moreover, statistical analysis tools may be used to address spatiotemporal variation in global
sampling efforts. New data from under-sampled areas such as the South Pacific will likely lead to new species discoveries and





may greatly improve the global observational basis of phytoplankton occurrence data in the future. Data inclusion from recent cruises, which are still under evaluation, appears as a natural next step. These data may come from the Malaspina expedition (Duarte, 2015), TARA Oceans (Bork et al., 2015) and transects in the Southern Ocean (Balch et al., 2016).

## 5 Data availability

PhytoBase is publicly available through PANGAEA, doi:10.1594/PANGAEA.904397 (Righetti et al., 2019a). Associated R scripts and the synonym tables used to harmonize species' names may be requested from the authors.

## 6 Conclusions

In PhytoBase, we compiled more than 1.35 million marine phytoplankton records that span 1716 species and nine major taxa or groups, including *Bacillariophyceae*, *Dinoflagellata, Haptophyta*, *Cyanobacteria* and others. The database addresses

photosynthetic microbial organisms, which play crucial roles in global biogeochemical cycles and marine ecology. We have provided an analysis of the current status of marine phytoplankton occurrence records accessible through public archives, their spatial and methodological limitations, and the completeness of species richness information for different ocean regions. PhytoBase may stimulate studies on the biogeography, diversity, and composition of phytoplankton and serve to calibrate ecological or mechanistic models. We recommend accounting carefully for data structure and metadata, depending on the

purpose of analysis.

## 7 Appendices

**Table A1:** Harmonization of 113 taxon names in the MareDat dataset of O'Brien et al. (2013). Only the 113 names that changed during harmonization are shown, out of a total of 197 names.

| Group | Original name | Harmonized name |
|---|---|---|
| *Haptophyta* | *_P. pouchetii* | *Phaeocystis pouchetii* |
| | *_P. pouchetii_* | *Phaeocystis pouchetii* |
| | *_Phaeocystis pouchetii* | *Phaeocystis pouchetii* |
| | *_Phaeocystis pouchetii (Subcomponent: bladders)* | *Phaeocystis pouchetii* |
| | *_Phaeocystis spp.* | *Phaeocystis* |
| | *_Phaeocystis spp._* | *Phaeocystis* |
| | *_Phaeocystis spp. (Subgroup: motile)* | *Phaeocystis* |
| | *_Phaeocystis spp. (Subgroup: non-motile)* | *Phaeocystis* |
| | *ACANTHOICA QUATTROSPINA* | *Acanthoica quattrospina* |
| | *Acanthoica acanthos* | *Anacanthoica acanthos* |
| | *Acanthoica sp. cf. quattraspina* | *Acanthoica quattrospina* |
| | *Algirosphaera oryza* | *Algirosphaera robusta* |





| | |
|---|---|
| *Algirosphaera robsta* | *Algirosphaera robusta* |
| *Anoplosolenia* | *Anoplosolenia brasiliensis* |
| *Anoplosolenia braziliensis* | *Anoplosolenia brasiliensis* |
| *Anoplosolenia sp. cf. brasiliensis* | *Anoplosolenia brasiliensis* |
| *Anthosphaera robusta* | *Algirosphaera robusta* |
| *CALCIDISCUS leptoporus* | *Calcidiscus leptoporus* |
| *Calcidiscus leptopora* | *Calcidiscus leptoporus* |
| *Calcidiscus leptoporus (inc. Coccolithus pelagicus)* | *Calcidiscus leptoporus* |
| *Calcidiscus leptoporus (small + intermediate)* | *Calcidiscus leptoporus* |
| *Calcidiscus leptoporus intermediate* | *Calcidiscus leptoporus* |
| *Calciosolenia MURRAYI* | *Calciosolenia murrayi* |
| *Calciosolenia brasiliensis* | *Anoplosolenia brasiliensis* |
| *Calciosolenia granii v closterium* | *Anoplosolenia brasiliensis* |
| *Calciosolenia granii v cylindrothecaf* | *Calciosolenia murrayi* |
| *Calciosolenia granii v cylindrothecaforma* | *Calciosolenia murrayi* |
| *Calciosolenia granii var closterium* | *Anoplosolenia brasiliensis* |
| *Calciosolenia granii var cylindrothecaeiformis* | *Calciosolenia murrayi* |
| *Calciosolenia murray* | *Calciosolenia murrayi* |
| *Calciosolenia siniosa* | *Calciosolenia murrayi* |
| *Calciosolenia sinuosa* | *Calciosolenia murrayi* |
| *Calciosolenia sp. cf. murrayi* | *Calciosolenia murrayi* |
| *Caneosphaera molischii* | *Syracosphaera molischii* |
| *Caneosphaera molischii and similar* | *Syracosphaera molischii* |
| *Coccolithus fragilis* | *Oolithotus fragilis* |
| *Coccolithus huxley* | *Emiliania huxleyi* |
| *Coccolithus huxleyi* | *Emiliania huxleyi* |
| *Coccolithus leptoporus* | *Calcidiscus leptoporus* |
| *Coccolithus sibogae* | *Umbilicosphaera sibogae* |
| *Crenalithus sessilis* | *Reticulofenestra sessilis* |
| *Crystallolithus cf rigidus* | *Calcidiscus leptoporus* |
| *Cyclococcolithus fragilis* | *Oolithotus fragilis* |
| *Discophaera tubifer* | *Discosphaera tubifera* |
| *Discosphaera thomsoni* | *Discosphaera tubifera* |
| *Discosphaera tubifer* | *Discosphaera tubifera* |
| *Discosphaera tubifer (inc. Papposphaera.lepida)* | *Discosphaera tubifera* |
| *Discosphaera tubifera* | *Discosphaera tubifera* |
| *Emiliana huxleyi* | *Emiliania huxleyi* |
| *Emiliania huxleyi A1* | *Emiliania huxleyi* |
| *Emiliania huxleyi A2* | *Emiliania huxleyi* |
| *Emiliania huxleyi A3* | *Emiliania huxleyi* |
| *Emiliania huxleyi C* | *Emiliania huxleyi* |
| *Emiliania huxleyi Indet.* | *Emiliania huxleyi* |





| | |
|---|---|
| *Emiliania huxleyi var. Huxleyi* | *Emiliania huxleyi* |
| *Florisphaera profunda var. profunda* | *Florisphaera profunda* |
| *Halopappus adriaticus* | *Michaelsarsia adriaticus* |
| *Helicosphaera carteri var. Carteri* | *Helicosphaera carteri* |
| *Michelsarsia elegans* | *Michaelsarsia elegans* |
| *Oolithotus fragilis var. Fragilis* | *Oolithotus fragilis* |
| *Oolithus spp. cf fragilis* | *Oolithotus fragilis* |
| *Ophiaster hydroideuss* | *Ophiaster hydroideus* |
| *Ophiaster spp. cf. Hydroides* | *Ophiaster hydroideus* |
| *P. antarctica* | *Phaeocystis antarctica* |
| *P. antarctica_* | *Phaeocystis antarctica* |
| *PHAEOCYSTIS* | *Phaeocystis* |
| *PHAEOCYSTIS_* | *Phaeocystis* |
| *PHAEOCYSTIS POUCHETII* | *Phaeocystis pouchetii* |
| *PHAEOCYSTIS POUCHETII_* | *Phaeocystis pouchetii* |
| *PHAEOCYSTIS sp.* | *Phaeocystis* |
| *PHAEOCYSTIS sp._* | *Phaeocystis* |
| *Palusphaera sp.* | *Rhabdosphaera longistylis* |
| *Palusphaera vandeli* | *Rhabdosphaera longistylis* |
| *Phaeocystis antarctica_* | *Phaeocystis antarctica* |
| *Phaeocystis cf. pouchetii* | *Phaeocystis pouchetii* |
| *Phaeocystis cf. pouchetii_* | *Phaeocystis pouchetii* |
| *Phaeocystis globosa_* | *Phaeocystis globosa* |
| *Phaeocystis motile* | *Phaeocystis* |
| *Phaeocystis motile_* | *Phaeocystis* |
| *Phaeocystis sp.* | *Phaeocystis* |
| *Phaeocystis sp._* | *Phaeocystis* |
| *Phaeocystis spp.* | *Phaeocystis* |
| *Pontosphaera huxleyi* | *Emiliania huxleyi* |
| *Rhabdosphaera  sp. cf. claviger (inc. var. stylifera)* | *Rhabdosphaera clavigera* |
| *Rhabdosphaera claviger* | *Rhabdosphaera clavigera* |
| *Rhabdosphaera clavigera var. Clavigera* | *Rhabdosphaera clavigera* |
| *Rhabdosphaera clavigera var. Stylifera* | *Rhabdosphaera clavigera* |
| *Rhabdosphaera stylifera* | *Rhabdosphaera clavigera* |
| *Rhabdosphaera tubifer* | *Discosphaera tubifera* |
| *Rhabdosphaera tubulosa* | *Discosphaera tubifera* |
| *Syrachosphaera pulchra* | *Syracosphaera pulchra* |
| *Syracosphaera brasiliensis* | *Anoplosolenia brasiliensis* |
| *Syracosphaera cf. Pulchra* | *Syracosphaera pulchra* |
| *Syracosphaera confuse* | *Ophiaster hydroideus* |
| *Syracosphaera corii* | *Michaelsarsia adriaticus* |
| *Syracosphaera cornifera* | *Helladosphaera cornifera* |





| | | |
|---|---|---|
| | *Syracosphaera corri* | *Michaelsarsia adriaticus* |
| | *Syracosphaera mediterranea* | *Coronosphaera mediterranea* |
| | *Syracosphaera molischii s.l.* | *Syracosphaera molischii* |
| | *Syracosphaera oblonga* | *Calyptrosphaera oblonga* |
| | *Syracosphaera quadricornu* | *Algirosphaera robusta* |
| | *Syracosphaera sp. cf. prolongata (inc. S.pirus)* | *Syracosphaera prolongata* |
| | *Syracosphaera tuberculata* | *Coronosphaera mediterranea* |
| | *Umbellosphaera hulburtiana* | *Umbilicosphaera hulburtiana* |
| | *Umbellosphaera sibogae* | *Umbilicosphaera sibogae* |
| | *Umbellosphaera spp. cf. irregularis + tenuis* | *Umbellosphaera irregularis* |
| | *Umbilicosphaera mirabilis* | *Umbilicosphaera sibogae* |
| | *Umbilicosphaera sibogae (Weber-van-Bosse) Gaarder* | *Umbilicosphaera sibogae* |
| | *Umbilicosphaera sibogae sibogae* | *Umbilicosphaera sibogae* |
| | *Umbilicosphaera sibogae var. Sibogae* | *Umbilicosphaera sibogae* |
| | *Umbilicosphaera spp. (U.sibogae)* | *Umbilicosphaera sibogae* |
| | *Umbillicosphaera sibogae* | *Umbilicosphaera sibogae* |

Note. An empty space in the original taxon name is indicated by "_".

**Table A2:** Harmonization of 156 taxon names in the MareDat dataset of Leblanc et al. (2012). Only the 156 names that changed during harmonization are shown, out of a total of 248 names.

| Group | Original name | Harmonized name |
|---|---|---|
| *Bacillariophyceae* | *Actinocyclus coscinodiscoides* | *Roperia tesselata* |
| | *Actinocyclus tessellatus* | *Roperia tesselata* |
| | *Asterionella frauenfeldii* | *Thalassionema frauenfeldii* |
| | *Asterionella glacialis* | *Asterionellopsis glacialis* |
| | *Asterionella mediterranea subsp pacifica* | *Lioloma pacificum* |
| | *Asterionellopsis japonica* | *Asterionellopsis glacialis* |
| | *Bacteriastrum varians* | *Bacteriastrum furcatum* |
| | *Cerataulina bergonii* | *Cerataulina pelagica* |
| | *Cerataulus bergonii* | *Cerataulina pelagica* |
| | *Ceratoneis closterium* | *Cylindrotheca closterium* |
| | *Ceratoneis longissima* | *Nitzschia longissima* |
| | *Chaetoceros angulatus* | *Chaetoceros affinis* |
| | *Chaetoceros atlanticus f. bulosus* | *Chaetoceros bulbosus* |
| | *Chaetoceros audax* | *Chaetoceros atlanticus* |
| | *Chaetoceros borealis f. concavicornis* | *Chaetoceros concavicornis* |
| | *Chaetoceros cellulosus* | *Chaetoceros lorenzianus* |
| | *Chaetoceros chilensis* | *Chaetoceros peruvianus* |
| | *Chaetoceros contortus* | *Chaetoceros compressus* |
| | *Chaetoceros convexicornis* | *Chaetoceros peruvianus* |
| | *Chaetoceros dichaeta* | *Chaetoceros distans* |





| | |
|---|---|
| *Chaetoceros dispar* | *Chaetoceros atlanticus* |
| *Chaetoceros grunowii* | *Chaetoceros decipiens* |
| *Chaetoceros jahnischianus* | *Chaetoceros distans* |
| *Chaetoceros javanis* | *Chaetoceros affinis* |
| *Chaetoceros peruvio-atlanticus* | *Chaetoceros peruvianus* |
| *Chaetoceros polygonus* | *Chaetoceros atlanticus* |
| *Chaetoceros radians* | *Chaetoceros socialis* |
| *Chaetoceros radiculus* | *Chaetoceros bulbosus* |
| *Chaetoceros ralfsii* | *Chaetoceros affinis* |
| *Chaetoceros remotus* | *Chaetoceros distans* |
| *Chaetoceros schimperianus* | *Chaetoceros bulbosus* |
| *Chaetoceros schuttii* | *Chaetoceros affinis* |
| *Chaetocros vermiculatus* | *Chaetoceros debilis* |
| *Corethron criophilum* | *Corethron pennatum* |
| *Corethron hystrix* | *Corethron pennatum* |
| *Corethron valdivae* | *Corethron pennatum* |
| *Coscinodiscus anguste-lineatus* | *Thalassiosira anguste-lineata* |
| *Coscinodiscus gravidus* | *Thalassiosira gravida* |
| *Coscinodiscus pelagicus* | *Thalassiosira gravida* |
| *Coscinodiscus polychordus* | *Thalassiosira anguste-lineata* |
| *Coscinodiscus rotulus* | *Thalassiosira gravida* |
| *Coscinodiscus sol* | *Planktoniella sol* |
| *Coscinodiscus sublineatus* | *Thalassiosira anguste-lineata* |
| *Coscinosira polychordata* | *Thalassiosira anguste-lineata* |
| *Dactyliosolen mediterraneus* | *Leptocylindrus mediterraneus* |
| *Dactyliosolen meleagris* | *Leptocylindrus mediterraneus* |
| *Detonula delicatula* | *Detonula pumila* |
| *Diatoma rhombica* | *Fragilariopsis rhombica* |
| *Dicladia bulbosa* | *Chaetoceros bulbosus* |
| *Dithylim inaequale* | *Ditylum brightwellii* |
| *Dithylum trigonum* | *Ditylum brightwellii* |
| *Eucampia balaustium* | *Eucampia antarctica* |
| *Eucampia Britannica* | *Eucampia zodiacus* |
| *Eucampia nodosa* | *Eucampia zodiacus* |
| *Eucampia striata* | *Guinardia striata* |
| *Eupodiscus tesselatus* | *Roperia tesselata* |
| *Fragilaria arctica* | *Fragilariopsis oceanica* |
| *Fragilaria kerguelensis* | *Fragilariopsis kerguelensis* |
| *Fragilaria obliquecostata* | *Fragilariopsis obliquecostata* |
| *Fragilaria rhombica* | *Fragilariopsis rhombica* |
| *Fragilariopsis antarctica* | *Fragilariopsis oceanica* |
| *Fragilariopsis sublinearis* | *Fragilariopsis obliquecostata* |





| | |
|---|---|
| *Fragilaris sublinearis* | *Fragilariopsis obliquecostata* |
| *Fragillariopsis antarctica* | *Fragilariopsis kerguelensis* |
| *Gallionella sulcata* | *Paralia sulcata* |
| *Guinardia baltica* | *Guinardia flaccida* |
| *Hemiaulus delicatulus* | *Hemiaulus hauckii* |
| *Henseniella baltica* | *Guinardia flaccida* |
| *Homeocladia closterium* | *Cylindrotheca closterium* |
| *Homeocladia delicatissima* | *Pseudo-nitzschia delicatissima* |
| *Lauderia borealis* | *Lauderia annulata* |
| *Lauderia pumila* | *Detonula pumila* |
| *Lauderia schroederi* | *Detonula pumila* |
| *Leptocylindrus belgicus* | *Leptocylindrus minimus* |
| *Melosira costata* | *Skeletonema costatum* |
| *Melosira marina* | *Paralia sulcata* |
| *Melosira sulcata* | *Paralia sulcata* |
| *Moerellia cornuta* | *Eucampia cornuta* |
| *Navicula mebranacea* | *Meuniera membranacea* |
| *Navicula planamembranacea* | *Ephemera planamembranacea* |
| *Navicula pseudomembranacea* | *Meuniera membranacea* |
| *Nitzschia actydrophila* | *Pseudo-nitzschia delicatissima* |
| *Nitzschia angulate* | *Fragilariopsis rhombica* |
| *Nitzschia Antarctica* | *Fragilariopsis rhombica* |
| *Nitzschia birostrata* | *Nitzschia longissima* |
| *Nitzschia closterium* | *Cylindrotheca closterium* |
| *Nitzschia curvirostris* | *Cylindrotheca closterium* |
| *Nitzschia delicatissima* | *Pseudo-nitzschia delicatissima* |
| *Nitzschia grunowii* | *Fragilariopsis oceanica* |
| *Nitzschia heimii* | *Pseudo-nitzschia heimii* |
| *Nitzschia kergelensis* | *Fragilariopsis kerguelensis* |
| *Nitzschia obliquecostata* | *Fragilariopsis obliquecostata* |
| *Nitzschia pungens* | *Pseudo-nitzschia pungens* |
| *Nitzschia seriata* | *Pseudo-nitzschia seriata* |
| *Nitzschiella longissima* | *Nitzschia longissima* |
| *Nitzschiella tenuirostris* | *Cylindrotheca closterium* |
| *Orthoseira angulate* | *Thalassiosira angulata* |
| *Orthoseira marina* | *Paralia sulcata* |
| *Orthosira marina* | *Paralia sulcata* |
| *Paralia marina* | *Paralia sulcata* |
| *Planktoniella wolterecki* | *Planktoniella sol* |
| *Podosira subtilis* | *Thalassiosira subtilis* |
| *Proboscia alata f. alata* | *Proboscia alata* |
| *Proboscia alata f. gracillima* | *Proboscia alata* |



| | |
|---|---|
| *Proboscia gracillima* | *Proboscia alata* |
| *Pyxilla baltica* | *Rhizosolenia setigera* |
| *Rhizosolenia alata* | *Proboscia alata* |
| *Rhizosolenia alata f. indica* | *Proboscia indica* |
| *Rhizosolenia alata var. indica* | *Proboscia indica* |
| *Rhizosolenia amputata* | *Rhizosolenia bergonii* |
| *Rhizosolenia antarctica* | *Guinardia cylindrus* |
| *Rhizosolenia calcar* | *Pseudosolenia calcar-avis* |
| *Rhizosolenia calcar avis* | *Pseudosolenia calcar-avis* |
| *Rhizosolenia calcar-avis* | *Pseudosolenia calcar-avis* |
| *Rhizosolenia cylindrus* | *Guinardia cylindrus* |
| *Rhizosolenia delicatula* | *Guinardia delicatula* |
| *Rhizosolenia flaccida* | *Guinardia flaccida* |
| *Rhizosolenia fragilima* | *Dactyliosolen fragilissimus* |
| *Rhizosolenia fragilissima* | *Dactyliosolen fragilissimus* |
| *Rhizosolenia genuine* | *Proboscia alata* |
| *Rhizosolenia gracillima* | *Proboscia alata* |
| *Rhizosolenia hebetata f hiemalis* | *Rhizosolenia hebetata* |
| *Rhizosolenia hebetata f. hebetata* | *Rhizosolenia hebetata* |
| *Rhizosolenia hebetata f. semispina* | *Rhizosolenia hebetata* |
| *Rhizosolenia hensenii* | *Rhizosolenia setigera* |
| *Rhizosolenia indica* | *Proboscia indica* |
| *Rhizosolenia japonica* | *Rhizosolenia setigera* |
| *Rhizosolenia murrayana* | *Rhizosolenia chunii* |
| *Rhizosolenia semispina* | *Rhizosolenia hebetata* |
| *Rhizosolenia stolterfothii* | *Guinardia striata* |
| *Rhizosolenia strubsolei* | *Rhizosolenia imbricata* |
| *Rhizosolenia styliformis var. longispina* | *Rhizosolenia styliformis* |
| *Rhizosolenia styliformis var. polydactyla* | *Rhizosolenia styliformis* |
| *Rhizosolenia styliformis var. semispina* | *Rhizosolenia hebetata* |
| *Schroederella delicatula* | *Detonula pumila* |
| *Spingeria bacillaris* | *Thalassionema bacillare* |
| *Stauroneis membranacea* | *Meuniera membranacea* |
| *Stauropsis membranacea* | *Meuniera membranacea* |
| *Synedra nitzschioides* | *Thalassionema nitzschioides* |
| *Synedra thalassiothrix* | *Thalassiothrix longissima* |
| *Terebraria kerguelensis* | *Fragilariopsis kerguelensis* |
| *Thalassionema elegans* | *Thalassionema bacillare* |
| *Thalassiosira condensata* | *Detonula pumila* |
| *Thalassiosira decipiens* | *Thalassiosira angulate* |
| *Thalassiosira polychorda* | *Thalassiosira anguste-lineata* |
| *Thalassiosira rotula* | *Thalassiosira gravida* |





| | |
|---|---|
| *Thalassiosira tcherniai* | *Thalassiosira gravida* |
| *Thalassiothrix curvata* | *Thalassionema nitzschioides* |
| *Thalassiothrix delicatula* | *Lioloma delicatulum* |
| *Thalassiothrix frauenfeldii* | *Thalassionema frauenfeldii* |
| *Thalassiothrix fraunfeldii* | *Thalassionema nitzschioides* |
| *Thalassiothrix mediterranea var. pacifica* | *Lioloma pacificum* |
| *Trachysphenia australis v kerguelensis* | *Fragilariopsis kerguelensis* |
| *Triceratium brightwellii* | *Ditylum brightwellii* |
| *Zygoceros pelagica* | *Cerataulina pelagica* |
| *Zygoceros pelagicum* | *Cerataulina pelagica* |

**Table A3:** Harmonization of the total of 109 species names in the data from Villar et al. (2015). Only the 109 names that changed during harmonization are shown, out of a total of 201 names.

| Group | Original name | Harmonized name |
|---|---|---|
| *Bacillariophyceae* | *Asteromphalus cf. flabellatus* | *Asteromphalus* |
| | *Asteromphalus spp.* | *Asteromphalus* |
| | *Bacteriastrum cf. delicatulum* | *Bacteriastrum* |
| | *Bacteriastrum cf. elongatum* | *Bacteriastrum* |
| | *Bacteriastrum cf. furcatum* | *Bacteriastrum* |
| | *Bacteriastrum cf. hyalinum* | *Bacteriastrum* |
| | *Bacteriastrum spp.* | *Bacteriastrum* |
| | *Biddulphia spp.* | *Biddulphia* |
| | *Chaetoceros atlanticus var. neapolitanus* | *Chaetoceros atlanticus* |
| | *Chaetoceros bulbosum* | *Chaetoceros bulbosus* |
| | *Chaetoceros cf. atlanticus* | *Chaetoceros* |
| | *Chaetoceros cf. coarctatus* | *Chaetoceros* |
| | *Chaetoceros cf. compressus* | *Chaetoceros* |
| | *Chaetoceros cf. danicus* | *Chaetoceros* |
| | *Chaetoceros cf. densus* | *Chaetoceros* |
| | *Chaetoceros cf. dichaeta* | *Chaetoceros* |
| | *Chaetoceros cf. laciniosus* | *Chaetoceros* |
| | *Chaetoceros cf. lorenzianus* | *Chaetoceros* |
| | *Chaetoceros spp.* | *Chaetoceros* |
| | *Climacodium cf. fravenfeldianum* | *Climacodium* |
| | *Climacodium spp.* | *Climacodium* |
| | *Corethron cf. pennatum* | *Corethron* |
| | *Corethron spp.* | *Corethron* |
| | *Coscinodiscus spp.* | *Coscinodiscus* |
| | *Cylindrotheca spp.* | *Cylindrotheca* |
| | *Ditylum spp.* | *Ditylum* |





| | | |
|---|---|---|
| | *Eucampia antartica* | *Eucampia antarctica* |
| | *Eucampia spp.* | *Eucampia* |
| | *Eucampia zodiacus f. cylindrocornis* | *Eucampia zodiacus* |
| | *Fragilariopsis spp.* | *Fragilariopsis* |
| | *Haslea wawrickae* | *Haslea wawrikae* |
| | *Hemiaulus spp.* | *Hemiaulus* |
| | *Hemidiscus cf. cuneiformis* | *Hemidiscus* |
| | *Lauderia spp.* | *Lauderia* |
| | *Leptocylindrus cf. danicus* | *Leptocylindrus* |
| | *Leptocylindrus cf. minimus* | *Leptocylindrus* |
| | *Lithodesmium spp.* | *Lithodesmium* |
| | *Nitzschia spp.* | *Nitzschia* |
| | *Odontella spp.* | *Odontella* |
| | *Pseudo-nitzschia cf. fraudulenta* | *Pseudo-nitzschia* |
| | *Pseudo-nitzschia cf. subcurvata* | *Pseudo-nitzschia* |
| | *Pseudo-nitzschia delicatissima group* | *Pseudo-nitzschia delicatissima* |
| | *Pseudo-nitzschia pseudodelicatissima group* | *Pseudo-nitzschia pseudodelicatissima* |
| | *Pseudo-nitzschia seriata group* | *Pseudo-nitzschia seriata* |
| | *Pseudo-nitzschia spp.* | *Pseudo-nitzschia* |
| | *Rhizosolenia cf. acuminata* | *Rhizosolenia* |
| | *Rhizosolenia cf. bergonii* | *Rhizosolenia* |
| | *Rhizosolenia cf. curvata* | *Rhizosolenia* |
| | *Rhizosolenia cf. decipiens* | *Rhizosolenia* |
| | *Rhizosolenia cf. hebetata* | *Rhizosolenia* |
| | *Rhizosolenia cf. imbricata* | *Rhizosolenia* |
| | *Rhizosolenia spp.* | *Rhizosolenia* |
| | *Skeletonema spp.* | *Skeletonema* |
| | *Thalassionema spp.* | *Thalassionema* |
| | *Thalassiosira spp.* | *Thalassiosira* |
| *Dinoflagellata* | *Amphidinium spp.* | *Amphidinium* |
| | *Archaeperidinium cf. minutum* | *Archaeperidinium* |
| | *Blepharocysta spp.* | *Blepharocysta* |
| | *Ceratocorys cf. gourreti* | *Ceratocorys* |
| | *Ceratocorys spp.* | *Ceratocorys* |
| | *Dinophysis cf. acuminata* | *Dinophysis* |
| | *Dinophysis cf. ovum* | *Dinophysis* |
| | *Dinophysis cf. uracantha* | *Dinophysis* |
| | *Dinophysis spp.* | *Dinophysis* |
| | *Diplopsalis group* | *Diplopsalis* |
| | *Gonyaulax cf. apiculata* | *Gonyaulax* |
| | *Gonyaulax cf. elegans* | *Gonyaulax* |
| | *Gonyaulax cf. fragilis* | *Gonyaulax* |





| | |
|---|---|
| *Gonyaulax cf. hyalina* | *Gonyaulax* |
| *Gonyaulax cf. pacifica* | *Gonyaulax* |
| *Gonyaulax cf. polygramma* | *Gonyaulax* |
| *Gonyaulax cf. scrippsae* | *Gonyaulax* |
| *Gonyaulax cf. sphaeroidea* | *Gonyaulax* |
| *Gonyaulax cf. spinifera* | *Gonyaulax* |
| *Gonyaulax cf. striata* | *Gonyaulax* |
| *Gonyaulax spp.* | *Gonyaulax* |
| *Gymnodinium spp.* | *Gymnodinium* |
| *Gyrodinium spp.* | *Gyrodinium* |
| *Histioneis cf. megalocopa* | *Histioneis* |
| *Histioneis cf. striata* | *Histioneis* |
| *Oxytoxum cf. laticeps* | *Oxytoxum* |
| *Oxytoxum spp.* | *Oxytoxum* |
| *Paleophalacroma unicinctum* | *Palaeophalacroma unicinctum* |
| *Phalacroma cf. rotundatum* | *Phalacroma* |
| *Prorocentrum cf. balticum* | *Prorocentrum* |
| *Prorocentrum cf. concavum* | *Prorocentrum* |
| *Prorocentrum cf. nux* | *Prorocentrum* |
| *Protoceratium spinolosum* | *Protoceratium spinulosum* |
| *Protoperidinium cf. bipes* | *Protoperidinium* |
| *Protoperidinium cf. breve* | *Protoperidinium* |
| *Protoperidinium cf. crassipes* | *Protoperidinium* |
| *Protoperidinium cf. diabolum* | *Protoperidinium* |
| *Protoperidinium cf. divergens* | *Protoperidinium* |
| *Protoperidinium cf. globulus* | *Protoperidinium* |
| *Protoperidinium cf. grainii* | *Protoperidinium* |
| *Protoperidinium cf. leonis* | *Protoperidinium* |
| *Protoperidinium cf. monovelum* | *Protoperidinium* |
| *Protoperidinium cf. nudum* | *Protoperidinium* |
| *Protoperidinium cf. ovatum* | *Protoperidinium* |
| *Protoperidinium cf. ovum* | *Protoperidinium* |
| *Protoperidinium cf. pyriforme* | *Protoperidinium* |
| *Protoperidinium cf. quarnerense* | *Protoperidinium* |
| *Protoperidinium cf. steinii* | *Protoperidinium* |
| *Protoperidinium cf. variegatum* | *Protoperidinium* |
| *Protoperidinuim spp.* | *Protoperidinium* |
| *Schuettiella cf. mitra* | *Schuettiella* |
| *Tripos arietinum* | *Tripos arietinus* |
| *Tripos lineatus/pentagonus complex* | *Tripos lineatus* |
| *Tripos massiliense* | *Tripos massiliensis* |

Note. Data of genera (using the harmonized names) were excluded from the database.

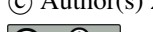



## 8 Author contributions

All authors contributed ideas to conceive the study. MV compiled substantial parts of the MareDat database. DR compiled the data, developed the code to perform the analyses, and wrote the initial manuscript, with substantial input by MV, NZ, and NG.

### Competing interests

The authors declare that they have no conflict of interest.

### Acknowledgements

We are grateful for expertise on taxonomic nomenclature provided by M. Guiry and M. Estrada. We thank all biologists, taxonomists and cruise organizers who spent tremendous efforts on collecting occurrences of marine phytoplankton species synthesized in this database. We thank M. Döring and P. Provoost for the help with retrieval of phytoplankton occurrence data from GBIF (www.gbif.org) and OBIS (www.obis.org). Funding for this effort came from ETH Zürich under grant ETH-52 13-2.

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
