# Peer review of "PHYTOBASE: A global synthesis of open ocean phytoplankton occurrences"

_Earth System Science Data, 2019_

## Referee Comment (RC1) · Griet NEUKERMANS (Referee) · 29 Oct 2019

This paper presents PhytoBase, a global dataset that is essentially a compilation of the existing GBIF and OBIS phytoplankton species occurrence datasets, and a few other smaller datasets. The synthesis and harmonization of these databases results in a substantial increase in phytoplankton occurrence records and yields the largest global database of phytoplankton occurrences. The PhytoBase dataset of spatiotemporal observations of species occurrences may contribute to studies that determine and forecast species distributions and studies aimed at understanding the drivers behind the distribution patterns. The limitations of the database are the spatially highly uneven data density, and, more importantly, strong biases due to differences in sampling methods (e.g. sampling volume, taxonomic resolution etc.). These limitations,

appropriately addressed in the paper, prevent the use of PhytoBase for direct analyses of species diversity patterns and biogeography studies, and severely limit the accuracy of data analyses. The authors thus correctly advise that statistical techniques be used to overcome the various biases present in PhytoBase.

I recommend publication of this database in ESSD. I only have a few very minor comments: i) What is the difference between Columns 1-2 and 3-4 in Tables 2 and 3? ii) Can you add a colorbar for the frequency distribution in Figure 3? iii)

---

## Referee Comment (RC2) · Anonymous Referee #2 · 12 Dec 2019

**Paper review**

**"PHYTOBASE: A global synthesis of open ocean phytoplankton occurrences"**

**Righetti et al.**

**REVIEW SUMMARY**

The authors present a compilation, named PhytoBase, of five data sources on phytoplankton occurrence records targeting open ocean, including two main data sources: the Global Biodiversity Information Facility (GBIF; www.gbif.org ), and the Ocean Biogeographic Information System (OBIS; www.obis.org), complemented by three other sources: the Marine Ecosystem Data initiative (MareDat; Buitenhuis et al. 2013), a marine micro-phytoplankton dataset (Sal et al., 2013), and with a subset of the data collected during the TARA Oceans cruise (Villar et al., 2015). To my knowledge, this compilation leads to the largest dataset on open ocean phytoplankton. A huge effort of data harmonization is recognized, on several aspects of both data structure and taxonomy but also on data qualification (cleaning) required to ensure data quality. This database opens perspectives for phytoplankton research on niche modelling, species distributions, especially within the context of global changes. This database, if updated and maintained in time, will be a valuable bibliographic source for future phytoplankton studies.

I would suggest the acceptance of the paper with 'Minor changes', but I give some recommendations that if addressed will contribute to strengthen and improve the quality of the present paper but also PhytoBase, in particular on data structuring and processes to maintain the valuable product. In my vision, the potential of PhytoBase lies in the compilation of existing data sources but essentially lies in its reproducibility, sustainability and maintenance in time instead of one single snapshot, even more because PhytoBase relies on existing data sources that are maintained and grow in time. That is the reason why some comments insist on sustainability and maintenance aspects, with emphasis on the need of reproducible processes.

**DETAILED COMMENTS**

**Abstract**

*No comments*

**1. Introduction**

*No comments*

**2. Compilation of occurrences**

**2.1. Data origin**
- Line 100-104: The authors should argue further why they have chosen these three complementary data sources in particular: the MareDat, data from Sal et al. and TARA

data collection subset. Did the authors proceed to some extensive bibliographic work to search for potentially valuable datasets and were the sources used the only available open datasets? If yes, this deserves further statements on this bibliographic work, and eventual criteria (if applicable) to choose the data sources.

- Lines 120-121: R packages *RPostgreSQL* and *devtools* should be properly cited and referenced

**2.2. Data selection**

**2.2.1. Data accessed through GBIF and OBIS**

- Lines 146-149: please provide percentage of records excluded with filters on year and missing date.

**2.3. Concatenation of source datasets**

- Line 188, In table 1: The authors do not mention in the main document that the two main data sources GBIF and OBIS extensively rely on the Darwin Core standard (https://dwc.tdwg.org/terms/). This explains why most of column names are the same. In addition, the authors should precise that an attempt was done to match Darwin Core standard in the final column names, as working to comply with a standard is in general a best practice and an added value for the work. Due to the fact the two main sources are aligned on Darwin Core, not mentioning Darwin Core might be seen as regression. To understand that the Darwin Core standard has been exploited by the authors, we have to refer to the CSV table (http://hs.pangaea.de/Projects/PHYTOBASE/Column_definition_for_phytoplankton_harmonized_database.zip ) available under section 'further details' of PhytoBase PANGEA record available at https://doi.pangaea.de/10.1594/PANGAEA.904397

- Lines 188, In table 1 : The table intends to harmonize the column names, and tries to use Darwin Core when possible. This is achieved with the following fields: scientificName, basisOfRecord, decimalLongitude, decimalLatitude, taxonRank, individualCount, year, month, day, but not for other fields. Indeed, for fields that are common to at least two data sources, such as Darwin Core column names for GBIF and OBIS, the table results in a some kind of de-harmonization and de-standardization of column names, such as for:

  o institutionCode (Darwin Core term), that is split in two separate columns specific to GBIF/OBIS, ie institutionCode_gbif / institutionCode_obis. It would have been preferable to keep the standard column name, and act at content level to keep source provenance, for example adding a prefix or URN such as *urn:gbif:* or *urn:obis:* as content of a single institutionCode

  o cellsPerLitre : This is a non standard term. It is recommended to keep aligning on the Darwin Core standard by relying on columns relative to measurements or facts, in particular to use standard terms measurementType ("number of cells"), measurementValue (value of cell number), and measurementUnit (number of cells per litre)

- o Depth: This is an non standard term and it deserves a reflexion whether the use of standard terms minimumDepthInMeters and maximumDepthInMeters could be relevant.

  In similar way, it is recommended that authors check in depth about existence of Darwin core terms that match the other column names: originDatabase, datasetKey, collectionCode, resname, resourceID, cruiseOrStationId, cruise, sampleId; while avoiding the use of source-specific column names, as illustrated above with the *institutionCode*.

- Lines 188, In table 1, row about cellsPerLitre: The authors should also revise the corresponding table row as it seems information has been wrongly copied-pasted ("taxonRank")

- Line 210. The authors make use of a column "group" to add either the Phylum or Class. It is recommended to keep using Darwin Core standard terms phylum and class as separate columns.

- Line 212: The authors make use of a column "sourceArchive" to refer to the data source from which the record comes from. It is recommended to look carefully at Darwin Core standard to find the appropriate standard term to use for referencing the data source.

- Beyond the harmonization of column names highlighted in Table 1, since I believe it is the core of paper describing the set-up of PhytoBase, it would be highly valuable to include in the main document the final data structure retained in the PhytoBase (as set in table http://hs.pangaea.de/Projects/PHYTOBASE/Column_definition_for_phytoplankton_harmonized_database.zip), including column definitions, and for the extra columns added by authors, to proceed with an in-depth check about existence of Darwin core terms to use instead of adhoc column names, as recommended with column names enumerated above. In fact, the high potential of PhytoBase and perspective to exploit it will be fostered by such Darwin Core standard compliance. By relying on Darwin Core, this will offer perspectives to facilitate growing of source global information systems such as GBIF or OBIS with datasets not yet available through it, while benefiting from data already harmonized and standardized through PhytoBase.

**2.3.1. Extant species selection and taxonomic harmonization**

- Lines 223-227: The authors refer to a screaning process performed by Algaebase founder and director, as personal communication. This screening led to exclude a relatively significant number of taxa and associated data. Hence, such process seems to appear as key harmonization task for PhytoBase. In my opinion, such process should be further described in the actual PhytoBase and paper materials & methods. In addition, there is no statement that make understand whether the screaning process was done manually or through a semi-automated procedure. If it is a manual process, this may be seen as a limitation referring to reproducibility, sustainability and maintenance of PhytoBase, even more because it has not been operated by PhytoBase creators/maintainers. It is then strongly recommended to describe further such

screening process within the main document (or through an appendix), and, if done manually, to suggest how this could be replaced or at least complemented by a semi-automated and reproducible process , thus leading to the possibility for future users to get an updated PhytoBase in time.

**2.3.2. Data merger and synthesis**

- Line 270: The *rgbif* R package should be properly cited and referenced. In addition, please note that there is a typo with the package name ('rgibf' instead of *rgbif*).

**3. Results**
**3.1. Data**
- This section is very welcome and acknowledged.

**3.1.1. Spatiotemporal coverage**

- Line 283: It is recommended to add the EPSG code of the World Geodetic System (WGS84). In addition, I recommend to include this as standard Darwin Core column in PhytoBase using the term geodeticDatum.

**5. Data availability**
- In principle, it is highly recommended, based on principles of open and reproducible science and sustainability, that authors make available already the R scripts together with the PhytoBase on PANGAEA, and avoid provision on demand through emails to the authors.

---

## Referee Comment (RC3) · Anonymous Referee #3 · 16 Dec 2019

The MS entitled "PHYTOBASE: A global synthesis of open ocean phytoplankton occurrences" by Righetti et al. represents an interesting effort of combining major existing marine phytoplankton diversity information gathered by microscopy observation, discrimination, identification and, for some of them cells and colony counts, all over ocean systems around the Globe. The authors take into account not only abundance (quantitative) but also presence (qualitative) information in the same database, as well as different sampling methodologies which have an impact on the results obtained, considering bigger or smaller organisms (according to mesh/silk size discrimination and/or microscopy limitations), delicate or robust species (which will not be disrupted by mesh collection), rare or abundant species (depending on the volume of sample analysed). The description of the data as well as the combination methodology, quality control,

flagging and taxonomic relevance/correction of the datasets before and after merging them, are clear. The authors make it possible to address a more complete picture by providing a direct and easier access to current knowledge of phytoplankton distribution all over the oceanic realm, identifying properly the uneven distribution od sampling effort and, consequently, of biodiversity assessment or phytoplankton in large areas mainly identified in the Southern Hemisphere. Moreover, they made also an assessment of which are the taxa well known in comparison which the taxa relatively poorly known, mainly concerning small phytoplankton. Finally, they clearly demonstrate the new possibilities in developing ecological models and predictions on the distribution of phytoplankton taxa in open ocean systems.

I therefore recommend this MS to be published in Earth System Science Data after some small technical corrections (see below).

Some general considerations:

One issue to be reminded is that one cannot state for sure, even considering areas which have been well sampled for decades, that some species are not present in a precise area, mostly because, in the corresponding esxisting databases, studies combining different sampling approaches and, to some extent, also different approaches for considering either morphology, molecular or functional diversity, are scarce.

It remains important then to make this new database as informative as possible, not only concerning the correct nomenclature to be used (and a big effort for make old and new names was also carried out by the present work) but also by considering biases due to different sampling strategies (either nets or tows, Niskin bottles, continuous pumping at a considered depth). One recommendation would be to maintain taxonomic and phylogenetical research as a complement of routine monitoring efforts, providing more accurate consideration of rare species by considering higher sample volumes, concentration by different manners and, the most important, taxonomist expertise which, combined to molecular phylogeny, will certainly make it possible to extract more information rom metabarcoding and metagenomic approaches. Moreover, it is also important to consider also new automated approaches which would make it possible to extend the sampling effort on different platforms, addressing most of the time a most limited taxonomical resolution but recalling on functional diversity which, to some extent, would complete taxonomical information included in a marine phytoplankton global database.

Some details:

Page 3 line 74: "...onto a 270 $\mu$m silk roll..." as it is important to remind the particular sampling conditions of CPR.

Page 6 line 170; what about other essential metadata as "collection device" and "analytical tool" (type of microscope) and "volume analysed"? Would this information be available/included/easy to access? Page 16: Figure 5 caption: "...temperate seas...of Southern Hemisphere (E), cold seas ...of Southern Hemisphere (F)..."

Page 18 lines 419-420: what about other biases of CPR collection as fragile unarmored species, small but also big as ciliates? An extra comment on this issue will be welcomed, as these surveys are one of the most sustained and complete surveys of plankton in some targeted areas.

Page 20 Figure 8 caption: References García et al. 2013; Locarinio et al., 2013 and de Boyer Montegut, 2004 are missing from the reference list.

Page 22 line 500: To what extent DNA sequencing have really become an alternative to microscopy for characterizing phytoplankton biogeography instead of a complementary and, to some extent supplementary to morphological microscopic identification?

Page 23 line 535: to what extent have you only considered photosynthetical microbial organisms only, especially in some major taxa where both heterotrophs and pigmented cells (mixotrophs or autotrophs) occur? Thanks for precising this in the Materials and Methods section.

---

## Author Response (AR1)

**Revised Manuscript Materials essd-2019-159**

**PhytoBase: A global synthesis of open ocean phytoplankton occurrences**

Damiano Righetti\*, Meike Vogt, Niklaus E. Zimmermann, Michael D. Guiry, Nicolas Gruber

\*Corresponding author. Email: damiano.righetti@env.ethz.ch

**This PDF file includes: Response to Reviews Marked copy of revised manuscript (track changes) Revised manuscript (clean version)**

In addition: all quality figures and the word document can be accessed via https://polybox.ethz.ch/index.php/s/e78FVPd6Gn8NJeS

**Summary by Righetti et al. (DR)**

We thank the three reviewers for their constructive comments, which provided a valuable basis for increasing the quality, reproducibility, and accuracy of the database and ms. In essence, reviewer 1 advised us to implement minor specifications in three data items. Reviewer 2 suggested minor changes with respect to the data structuring and methodology, with a main focus on facilitating future updates of our database and its curation over time. Reviewer 3 suggested a set of general discussion points and minor specifications.

We address each of these points in detail. Red markings indicate textual edits that have been implemented in the revised version of the manuscript. All statistics and figures in the manuscript have been thoroughly updated.

**Reviewer 1**:**

This paper presents PhytoBase, a global dataset that is essentially a compilation of the existing GBIF and OBIS phytoplankton species occurrence datasets, and a few other smaller datasets. The synthesis and harmonization of these databases results in a substantial increase in phytoplankton occurrence records and yields the largest global database of phytoplankton occurrences. The PhytoBase dataset of spatiotemporal observations of species occurrences may contribute to studies that determine and forecast species distributions and studies aimed at understanding the drivers behind the distribution patterns. The limitations of the database are the spatially highly uneven data density, and, more importantly, strong biases due to differences in sampling methods (e.g. sampling volume, taxonomic resolution etc.). These limitations, appropriately addressed in the paper, prevent the use of PhytoBase for direct analyses of species diversity patterns and biogeography studies, and severely limit the accuracy of data analyses. The authors thus correctly advise that statistical techniques be used to overcome the various biases present in PhytoBase.

I recommend publication of this database in ESSD. I only have a few very minor comments: i) What is the difference between Columns 1-2 and 3-4 in Tables 2 and 3? ii) Can you add a colorbar for the frequency distribution in Figure 3?

**Specific responses by Righetti et al (DR) to Reviewer 1 (RE1):**

DR: We thank RE1 for the careful check of our data items and change these as follows:
i) Line 227ff and 266ff: Using intersected lines, we now highlight that the first two columns summarize the total records, while the third and fourth column summarize the subset of records with a depth-statement. This distinction is important, as plankton compositions often shift with depth and analyses may thus focus on records with a depth that can be associated to the well-mixed upper water column (mixed layer depth).
ii) Line 313ff: We have added grey bars to each panel in Fig. 3 and specified the caption.

**Reviewer 2:**

**Review summary**

The authors present a compilation, named PhytoBase, of five data sources on phytoplankton occurrence records targeting open ocean, including two main data sources: the Global Biodiversity Information Facility (GBIF; www.gbif.org ), and the Ocean Biogeographic Information System (OBIS; www.obis.org), complemented by three other sources: the Marine Ecosystem Data initiative (MareDat; Buitenhuis et al. 2013), a marine micro-phytoplankton dataset (Sal et al., 2013), and with a subset of the data collected during the TARA Oceans cruise (Villar et al., 2015). To my knowledge, this compilation leads to the largest dataset on open ocean phytoplankton. A huge effort of data harmonization is recognized, on several aspects of both data structure and taxonomy but also on data qualification (cleaning) required to ensure data quality. This database opens perspectives for phytoplankton research on niche modelling, species distributions, especially within the context of global changes. This database, if updated and maintained in time, will be a valuable bibliographic source for future phytoplankton studies.

I would suggest the acceptance of the paper with 'Minor changes', but I give some recommendations that if addressed will contribute to strengthen and improve the quality of the present paper but also PhytoBase, in particular on data structuring and processes to maintain the valuable product. In my vision, the potential of PhytoBase lies in the compilation of existing data sources but essentially lies in its reproducibility, sustainability and maintenance in time instead of one single snapshot, even more because PhytoBase relies on existing data sources that are maintained and grow in time. That is the reason why some comments insist on sustainability and maintenance aspects, with emphasis on the need of reproducible processes.

**Interpretation of the aspects raised by Reviewer 2 (RE2):**

**DR:** We thank RE2 for the thorough analysis and constructive comments, which greatly improved the quality of our manuscript. We share every interest to facilitate future updates of PhytoBase. To ensure this "dynamic component", we increase transparency and clarity on our methods, in particular with regard to synthesizing original data and columns across sources (textual edits, see lines indicated below), and we now publish the 21 relevant R-scripts used to do download, clean, and synthesize PhytoBase on gitlab (https://gitlab.ethz.ch/phytobase/supplementary). In addition, we now publish the "synonymy table" on gitlab, which lists the original 3303 species names (or generic names) in the raw data together with the harmonized species names (or generic names). Line 540ff: "PhytoBase is publicly available through PANGAEA, doi:10.1594/PANGAEA.904397 (Righetti et al., 2019a). Associated R scripts and the synonymy table used to harmonize species' names are available through https://gitlab.ethz.ch/phytobase/supplementary.

**Detailed comments**

**Abstract No comments**

**1. Introduction No comments**

**2. Compilation of occurrences**

**2.1.Data origin**

- Line 100-104: The authors should argue further why they have chosen these three complementary data sources in particular: the MareDat, data from Sal et al. and TARA data collection subset. Did the authors proceed to some extensive bibliographic work to search for potentially valuable datasets and were the sources used the only available open datasets? If yes, this deserves further statements on this bibliographic work, and eventual criteria (if applicable) to choose the data sources.

**Specific responses by Righetti et al (DR):**

DR: We clarify our initial choice of data sources: The primary focus was set on retrieving data from GBIF (www.gbif.org) and OBIS (www.obis.org); firstly, because GBIF and OBIS promised the largest gain of data-points, as a function of time and effort spent (GBIF: 790'103 data points for 1492 species, with 54.9% of points being unique to this source; OBIS: 823'836 data points for 1320 species, with 56.3% of points being unique). Second, we focused on GBIF and OBIS, because a framework including these two growing archives, will ensure an efficient gathering of phytoplankton data in the future, in line with the mission statement of GBIF ("GBIF (...) is aimed at providing anyone, anywhere, open access to data about all types of life on Earth"; https://www.gbif.org/what-is-gbif, accessed 27.02.2020) and OBIS ("Vision: To be the most comprehensive gateway to the world's ocean biodiversity and biogeographic data (...)"; https://www.obis.org/about/, accessed 27.02.2020). Due to their strive for completeness, we expect OBIS and GBIF to remain leading archives for sharing biological data between multiple datasets and sources, and will serve themselves as key attractors for future datasets from various sources, including datasets from TARA Oceans, the MALASPINA expedition, and other marine diversity efforts. In this context, it will be the key task of individual institutions and cruises to inject their data into these two archives, rather than spreading data across multiple repositories, and to reconcile taxonomy with reference standards. Our work demonstrates how data can be efficiently inter-compared and merged between major plankton data archives.

Our choice of the three additional sources was, indeed, not exhaustive. It included a large dataset that was acquired, quality-controlled and published by our group, the MAREDAT data set, which we are highly familiar with (e.g., O'Brien et al., 2016; Brun et al., 2015; MAREDAT: 101'969 records, among which 94.7% were new to PhytoBase). We also strived to include data from the global TARA Oceans cruise, yet at the time of data download (closing window, March 2017) not all data from TARA Oceans were publicly available, and we thus limited the inclusion to the quality-controlled dataset of Villar et al. (2015). Last but not least, we added the global dataset from the AMT data series by Sal et al. (2013), which is unique in aspects of taxonomic standardization and consistency in sampling methodology. The inclusion of other, smaller datasets was beyond the scope of this project.

We thoroughly specify the selection of data in the revised version of the manuscript: Line 100ff: "To create PhytoBase, we compiled marine phytoplankton occurrences (i.e., presences or abundances) from five sources, including the two largest open-access species occurrence archives: the Global Biodiversity Information Facility (GBIF; www.gbif.org), and the Ocean Biogeographic Information System (OBIS; www.obis.org). These two archives represent leading efforts to globally gather species distribution evidence. We augmented the data with records from the Marine Ecosystem Data initiative (MareDat; Buitenhuis et al. 2013), records from a micro-phytoplankton dataset (Sal et al., 2013), and records from the global TARA Oceans cruise (Villar et al., 2015), which were not included in GBIF or OBIS at the time of data query (closing window, March 2017). While our selection of additional data was not exhaustive, it strived for the inclusion of quality controlled large-scale phytoplankton datasets. Specifically, MareDat represents a previous global effort in gathering marine plankton data for ecological analyses (e.g., Brun et al., 2015; O'Brien et al., 2016), while Sal et al. (2013) and Villar et al. (2015) are unique in aspects of taxonomic standardization and consistency in methodology."

DR: To avoid redundancies and increase clarity, we specify the subsequent sections: Lines 132ff : "(...). Occurrence data from the TARA Ocean cruise included the Bacillariophyceae and Dinoflagellata (Villar et al., 2015; their Tables W8 and W9). Occurrence data from MareDat included five phytoplankton papers (Buitenhuis et al., 2012; Leblanc et al., 2012; Luo et al., 2012; O'Brien et al., 2013; Vogt et al., 2012). Additional data processed by the TARA Oceans or Malaspina expedition (Duarte, 2015) may provide valuable context for a future data synthesis, yet here we have focused on publicly available sources until March 2017. The raw sources that underpin GBIF and OBIS, and MAREDAT, represent decades to centuries of efforts spent in collecting phytoplankton data, including a substantial amount of data from the CPR program (Richardson et al., 2006). In addition, a large fraction of data from the AMT program (cruises 1 to 6) are represented in Sal et al. (2013)."

Line 483ff: "The harmonization of different archives striving to collect global species evidence, therefore substantially expanded the empirical basis of phytoplankton records."

Lines 120-121: R packages RPostgreSQL and devtools should be properly cited and referenced

**DR:** We agree and cite the packages. In addition we reference the package 'robis'. Line 124 ff: "The data from OBIS were first retrieved on 5 December 2015 using the R package robis (Provoost and Bosch, 2015) and the OBIS taxonomic backbone, accessed on 4 December 2015 via the R packages RPostgreSQL (Conway et al., 2015) and devtools (Wickham, H. and *Chang, 2015). Data were updated for the taxa selected on 6 March 2017 (using the OBIS taxonomic backbone, accessed on 6 March 2017 via the same R packages)."*

**Line 575 ff:**

"Provoost, P. and Bosch, S.: robis: R client for the OBIS API. R package version 0.1.5. https://cran.r-project.org/package=robis, 2015. Conway, J., Eddelbuettel, D., Nishiyama, T., Prayaga, S. K., Tiffin, N.: RPostgreSQL: R interface to the PostgreSQL database system. R package version 0.4.1. https://cran.rproject.org/package=RPostgreSQL, 2015.

Wickham, H. and Chang, W.: Devtools: Tools to make developing R packages easier. R package version 1.12.0. https://cran.r-project.org/package=devtools, 2015."

**2.2.Data selection**

2.2.1. Data accessed through GBIF and OBIS

- Lines 146-149: please provide percentage of records excluded with filters on year and missing date.

DR: We revisited our statistics and now present this information in the main text: Line 149ff: "To filter out raw data of presumably inferior quality, records from OBIS and GBIF were removed: (i) if their year of collection indicated >2017 or <1800 (excluding 110 records; <0.001% of raw data), (ii) if they had no indication on the year or month of collection (excluding 7.2% GBIF raw data and 0.9% OBIS raw data) or (iii) if they had geographic coordinates outside the range -180 to 180 for longitude and/or outside -90 to 90 for latitude. The latter criterion did not lead to any data exclusion, as (...)"

Line 154ff has now been adjusted and specified accordingly: "*Records with negative* recording depths (0% of GBIF and 6.6% of OBIS raw data) were flagged and changed to positive, assuming that their original sign was mistaken."

Line 171ff has now been adjusted accordingly: "(...) we flagged rather than excluded data with reported recording before year 1800 (564 records; values 6, 10 or 11) and unrealistic day entries (58 340 records; values -9 or -1)."

**2.3.Concatenation of source datasets**

Line 188, In table 1: The authors do not mention in the main document that the two main data sources GBIF and OBIS extensively rely on the Darwin Core standard (https://dwc.tdwg.org/terms/). This explains why most of column names are the same. In

addition, the authors should precise that an attempt was done to match Darwin Core standard in the final column names, as working to comply with a standard is in general a best practice and an added value for the work. Due to the fact the two main sources are aligned on Darwin Core, not mentioning Darwin Core might be seen as regression. To understand that the Darwin Core standard has been exploited by the authors, we have to refer to the CSV table

(http://hs.pangaea.de/Projects/PHYTOBASE/Column\_definition\_for\_phytoplankton\_ harmonized\_database.zip ) available under section 'further details' of PhytoBase PANGEA record available at https://doi.pangaea.de/10.1594/PANGAEA.904397

**DR:** We thank the reviewer for this point. We now explain our naming convention at the first instance in the main text, which aligns with Darwin Core (dwc) standard wherever possible. We now provide an overview on the full column structure contained in PhytoBase in Table1 and highlight the column names that are in line with dwc. Upon contacting the GBIF secretariat, we received an additional expert opinion on the possibility for alignment of our original column names in PhytoBase with dwc.

Line 187ff: "Columns match Darwin Core standard (https://dwc.tdwg.org) where original data structure could be reconciled with this standard, following GBIF and OBIS that widely rely on Darwin Core. Where critical metadata could not be assigned to Darwin Core, we use additional columns (e.g., columns ending in "gbif" present metadata from GBIF)." We highlight the column names in line with dwc by a "\*", adding a note to Table 1: Line 196: "\*Column names following Darwin Core standard (https://dwc.tdwg.org)." We adjust the table's header, line 192: "Table 1: Harmonization of original column names (data-fields) between data sources and final column name structure in PhytoBase" We shorten the main text: Line 147: "(...) assuming that the latter was based on observation (see Table 1 for an overview of the metadata retained).

Lines 188, In table 1 : The table intends to harmonize the column names, and tries to use Darwin Core when possible. This is achieved with the following fields: scientificName, basisOfRecord, decimalLongitude, decimalLatitude, taxonRank, individualCount, year, month, day, but not for other fields. Indeed, for fields that are common to at least two data sources, such as Darwin Core column names for GBIF and OBIS, the table results in a some kind of de-harmonization and de-standardization of column names, such as for:
o institutionCode (Darwin Core term), that is split in two separate columns specific to

GBIF/OBIS, ie institutionCode\_gbif / institutionCode\_obis. It would have been preferable to keep the standard column name, and act at content level to keep source provenance, for example adding a prefix or URN such as urn:gbif: or urn:obis: as content of a single institutionCode o cellsPerLitre : This is a non standard term. It is recommended to keep aligning on the Darwin Core standard by relying on columns relative to measurements or facts, in particular to use standard terms measurementType ("number of cells"), measurementValue (value of cell number), and measurementUnit (number of cells per litre) o Depth: This is an non standard term and it deserves a reflexion whether the use of standard terms minimumDepthInMeters and maximumDepthInMeters could be relevant. DR: We now present the full column structure of PhytoBase in Table 1. The column names align with our revised naming convention (see above, revised line 189ff). We mark all column names in Table 1 that are in line with dwc standard.

**DR:** InstitutionCode: Entries on the "InstitutionCode" of records stemming from both OBIS and GBIF have been identical. We hence could perfectly merge the columns institutionCode\_gbif and institutionCode\_obis to a single column named "InstitutionCode", in line with dwc.

**DR:** cellsPerLitre: In line with RE2, and upon contacting the GBIF secretariat, we now split this column into two dwc terms: "organismQuantity" (here, we present the values) and "organismQuantityType" (i.e., "number\_of\_cells\_per\_L").

**DR:** Depth: We carefully examined the benefit of including the minimum– and maximum depth statement. However, "MinimumDepthInMeters" and "MaximumDepthInMeters" were not available for original GBIF records. By contrast, 18.6% of GBIF raw records contained a statement on "DepthAccuracy". This is because GBIF sticks to the term "depth" (as differing from dwc) and the single matching term "depthAccuracy". Similarly, among the OBIS records, 21.6% contained a "depthAccuracy", and only marginally more records contained a MinimumDepthInMeter (25.7%) or minimumDepthInMeter (24.0%). To enhance compatibility between the two major source archives in PhytoBase, we hence stick to the term "depth" together with "depthAccuracy", in line with GBIF data conventions. We now elaborate this point in the main text.

Line 190 ff: "With regard to sampling depth, GBIF raw data contained the field

"depthAccuracy" (i.e., a non Darwin Core term; 18.6% of records with entries) while OBIS raw data contained the fields "depthprecision" (21.6% of data with entries),

"minimumDepthInMeters" (25.7% of data with entries) and "maximumDepthInMeters" (24.0% of data with entries), i.e., two Darwin Core terms. To enhance compatibility between GBIF and OBIS, we therefore used the column "depth", together with "depthAccuracy", and we integrated "depthprecision" into the latter column."

**DR:** We note that depth accuracy statements have not been present in the raw data of Maredat, Villar et al. (2015) or Sal et al. (2013). This is mainly because discrete samples at specific depths have been analyzed for phytoplankton abundance and taxonomic identity.

**RE2:** In similar way, it is recommended that authors check in depth about existence of Darwin core terms that match the other column names: originDatabase, datasetKey, collectionCode, resname, resourceID, cruiseOrStationId, cruise, sampleId; while avoiding the use of source-specific column names, as illustrated above with the *institutionCode*. **DR:** We agree, and check the remaining column names for compatibility with Darwin Core.

**DR:** "originDatabase\_maredat" refers uniquely to MareDat (original name: "Origin Database" or "Database", depending on the MareDat paper). This column presents acronyms of original databases to which records belonged inside MareDat. In line with our naming convention provided in lines 187ff of the revised ms (i.e., Darwin Core where possible, specific columns for other relevant metadata where needed) we stick to the current term.

**DR:** "datasetKey" is a non dwc term, inherent to GBIF terminology: A closely related dwc term would be "datasetID". We thus tested whether we can merge "datasetKey" (inherent to GBIF data) and "resourceID" (inherent to OBIS data) into the single column named "datasetID", without creating ambiguity to which original source (GBIF, OBIS, MAREDAT, Villar or Sal) merged entries in "resourceID" would belong. We find that for 26.1% of data in PhytoBase, this merger would lead to two entries for "resourceID" – one leading back to OBIS, one back to GBIF. This is because a substantial part of the records have origin in both GBIF and OBIS. To keep column entries slim and retain important metadata, traceable to OBIS and GBIF, we decide to stick to the current columns, in line with our naming convention. In addition, we find that there are many more datasetKeys (GBIF) than resourceIDs (OBIS). Hence, retaining the detail of resolution seems advantageous.

DR: In line with our naming convention, we retain "collectionCode\_obis", "resname\_obis",

"resourceID\_obis", "cruiseOrStationID\_maredat", "cruise\_sal", and "sampleID\_sal" as separate columns. These columns contain metadata at different levels of detail, reflecting data structure in underlying source archives. This original resolution is important for future data users, as it allows associating the records to different cruises or protocols, and thus potentially different methodologies used in phytoplankton collection.

DR: We have removed column "ID", which is not conform with dwc. However, we now add a note to Table 1 guide the reader/user with the potential creation of an occurrence ID: Line 195: "Each record in PhytoBase is uniquely identifiable by the occurrence ID: scientificName, decimalLongitude, decimalLatitude, year, month, day, depth"

- Lines 188, In table 1, row about cellsPerLitre: The authors should also revise the corresponding table row as it seems information has been wrongly copied-pasted ("taxonRank")

**DR:** Indeed. "taxonRank" has been deleted from the erroneous places in Table 1.

- Line 210. The authors make use of a column "group" to add either the Phylum or Class. It is recommended to keep using Darwin Core standard terms phylum and class as separate columns.

**DR:** We have added the columns "phylum" and "class" (dwc standard terms) to PhytoBase, and remove "group". We thouroughly checked higher order taxonomy and adjusted the ms accordingly. We add a note to Table 1 on the higher order taxonomic hierarchy: Line 198: "†Higher order taxonomy (phylum, class) follows OBIS (taxonomic backbone; retrieved 6 March 2017), which relies on the World Register of Marine Species (www.marinespecies.org)".

**DR:** We update Table 4 in the MS accordingly (Line 351ff), We change *Dinoflagellatae* to *Dinophyceae* throughout the ms, and we adjust/clarify the names of key taxa: Lines 15ff: "(...) spanning the principal groups of the diatoms, dinoflagellates, and haptophytes, as well as several other groups."

Line 112ff: "More specifically, within the Ochrophyta, we considered the classes Bacillariophyceae (diatoms), Chrysophyceae, Dictyochophyceae, Pelagophyceae and Raphidophyceae. Within the Myzozoa, we considered the class Dinophyceae (dinoflagellates)." Line 478ff: "This new database contains 1 360 621 records (1 280 103 records at the level of species), including 1716 species of seven phyla."

Line 380: "#Including one species of the syster class Pelagophyceae."

Lines 478ff: "This new database contains 1 360 621 records (1 280 103 records at the level of species), including 1716 species of seven phyla."

Lines 547ff: "In PhytoBase, we compiled more than 1.36 million marine phytoplankton records

**that span 1704 species <del>and ten major groups,</del> including the key taxa Bacillariophyceae, Dinophyceae, Haptophyta, Cyanobacteria and others."**

Line 212: The authors make use of a column "sourceArchive" to refer to the data source from which the record comes from. It is recommended to look carefully at Darwin Core standard to find the appropriate standard term to use for referencing the data source.
DR: We agree that standard terms are preferable. Our column "sourceArchive" is unique to the PhytoBase compilation, indicating from what original, large archive (GBIF, OBIS, MAREDAT, Villar, or Sal) each record stems. The associated column "yearOfDataAccess" presents the year, in which data were downloaded from archives. We find no suitable matchup terms in dwc system for these purposes, and stick to the current terms.

- Beyond the harmonization of column names highlighted in Table 1, since I believe it is the core of paper describing the set-up of PhytoBase, it would be highly valuable to include in the main document the final data structure retained in the PhytoBase (as set in table http://hs.pangaea.de/Projects/PHYTOBASE/Column\_definition\_for\_phytoplankton\_h armonized\_database.zip), including column definitions, and for the extra columns added by authors, to proceed with an in-depth check about existence of Darwin core terms to use instead of adhoc column names, as recommended with column names enumerated above. In fact, the high potential of PhytoBase and perspective to exploit it will be fostered by such Darwin Core standard compliance. By relying on Darwin Core, this will offer perspectives to facilitate growing of source global information systems such as GBIF or OBIS with datasets not yet available through it, while benefiting from data already harmonized and standardized through PhytoBase.

**DR:** We agree with RE2 that a comprehensive presentation of column names is desirable. We adjust Table 1 accordingly. We now also elucidate the content of many columns in the footnotes of Table 1. Yet, given space constraints, we describe each column and their content more thoroughly in the Excel sheet, which is presenting all columns (accompanying PhytoBase on Pangaea). Moreover, Table 1 has been annotated to indicate dwc terms. See also our discussion, to what degree we make columns compatible with dwc in our response to the RE2's general comment on "2.3. Concatenation of source datasets".

**DR:** We checked the compatibility of added columns with dwc:

Regarding "sourceArchive" and "yearOfDataAccess" we stick to the original terms, in accordance with our response to line 212 (RE2, see above). We now explain why we include

the two columns in the main text.

Line 208ff: "To indicate the source from which records were obtained (GBIF, OBIS, MAREDAT, VILLAR or SAL) and the year of data access, we added the columns "sourceArchive" and "yearOfDataAccess".

**DR:** Regarding "colonialFormCellsPerLitre": We now integrate the column "colonialFormCellsPerLitre" into the columns "organismQuantity" and "organismQuantityType", using "number\_of\_colonial\_ form\_cells\_per\_L" as the entry for the latter. To maintain source attribution we highlight that quantifications for "colonial type cells" stem from MAREDAT

Line 166: "Across all sources, data on colonial cells were uniquely provided by MareDat, (...)."

**DR:** Regarding "totalColonialorSingleCells\_or\_trichomes\_l": We remove this column, as it cannot be reconciled with dwc, while adding only very minor additional data to PhytoBase. To compensate for this exclusion, we refer to the additional data in the text. Line 166: "Across all sources, data on colonial cells could be uniquely accessed via MareDat (and additional count data on trichomes of genus Trichodesmium are available from Luo et al., 2012)."

**DR:** Regarding "recordWithinMLD\_clim" and "depthOriginal". Both columns cannot be reconciled with dwc. We remove the first column (presenting climatological reference data from de Boyer and Montegut, 2004) and leave it now up to the data user to define the mixed-layer depth (if required to select data). The second column ("depthOriginal") can be reconstructed via the column "depth" and a new column "flag" (below). We hence delete it.

**DR:** Regarding "unrealisticDayOrYear" and "basisPresumablySedimentary": We replace these columns by a quality flag column, termed "flag". We explain the purpose of this column to the reader in the main text.

Line 210ff: "Last, we added a quality flag column, termed "flag". This column denotes records with originally negative collection depth entries (N) (sect. 2.2.1), unrealistic day (D) or year (Y) entries (sect. 2.2.2), and/or records collected from sediment samples or traps (S), rather than seawater samples (sect. 2.3.2).

Line 273 ff: We flagged phytoplankton records from OBIS and GBIF in the database associated with surface sediment traps or sediment cores (denoted by an "S", in the flag column) (...)".

**DR:** Accordingly, we correct all column names, and their explanation in the excel sheet that accompanies PhytoBase on Pangea.

**DR:** Owing to the changes in column name structure, in line with the inputs by RE2, the following sentences or sub-clauses have been deleted from the manuscript: Line 164ff: The column "unrealisticDayOrYear" in PhytoBase indicates day or year entries, originally associated with MareDat. Data selected from MareDat were merged to a single dataset, containing the columns: "scientificName", "longitude", "latitude", "year", "month", "day", "group", "Origin Database", "Cruise or station ID", "basis", "depth", and "rank".

Line 203ff: We added the column "group" to the database, denoting to which phylum or class records belong: i.e., *Cyanobacteria, Bacillariophyceae, Chlorophyta, Chrysophyceae, Cryptophyta, Dinoflagellata, Euglenophyta, Haptophyta, Raphidophyceae* or picoeukaryotes, and the column "sourceArchive", indicating the source from which records were obtained (GBIF, OBIS, MAREDAT, VILLAR or SAL).

Line 251 ff: Furthermore, we added the column "yearOfDataAccess", indicating the year of data download (2015, 2017 or both) and the column "containedWithinMLD\_clim", which distinguishes records stemming from waters deeper than the oceanic mixed-layer (monthly climatology, de Boyer Montégut 2004) (11.5% of records) from those inside the mixed-layer.

Line 265 ff: "(...) this does not exclude the possibility that occurrence records of extant species in the GBIF and OBIS source datasets originated partially from sediment traps or sediment core samples<del>, rather than from seawater samples</del>."

**2.3.1. Extant species selection and taxonomic harmonization**

- Lines 223-227: The authors refer to a screening process performed by Algaebase founder and director, as personal communication. This screening led to exclude a relatively significant number of taxa and associated data. Hence, such process seems to appear as key harmonization task for PhytoBase. In my opinion, such process should be further described in the actual PhytoBase and paper materials & methods. In addition, there is no statement that make understand whether the screaning process was done manually or through a semi-automated procedure. If it is a manual process, this may be seen as a limitation referring to reproducibility, sustainability and maintenance of PhytoBase, even more because it has not been operated by PhytoBase creators/maintainers. It is then strongly recommended to describe further such screening process within the main document (or through an appendix), and, if done manually, to suggest how this could be replaced or at least complemented by a semi-automated and reproducible process, thus leading to the possibility for future users to get an updated PhytoBase in time.

DR: First, we provide the necessary basis that any updated (or different) method can be implemented to standardize or harmonize the species names in PhytoBase: Line 197: "We retain all original scientificName(s) and synonyms used in individual sources as additional columns with the format "scientificNameOriginal\_<source>" Line 257ff: "In particular, we retained the original taxonomic name(s) associated with each record in separate columns of the type "scientificNameOriginal\_<source>", which allows tracing back the harmonized name to its original name(s). Retaining original names ensures that future taxonomic changes or updated methods can be readily implemented."

**DR:** Second, we agree with RE2 that the harmonization procedure should be further specified, which has now been implemented as follows.

Line 223ff: "(ii) We extracted all scientific names (mostly at species level, including all synonyms and spelling variants) associated with at least one depth-referenced record from the raw database (Table 2). This resulted in 3302 names, which were validated in August 2017 against the 150 000+ specific and infraspecific names in Algaebase (www.algaebase.org), and matched using a relational database of current names and synonyms; orthography was made as compatible as possible with the International Code of Nomenclature (Turland et al., 2018), particularly in relation to the gender of specific epithets. Each name was verified by M. Guiry, the founder and director at Algaebase (M. Guiry, pers. comm.) in August 2017. This expert screening led to the exclusion of 459 names (...).

(iii) We excluded species (and their data) classified as "fossil only" or "fossil", based on Algaebase (accessed August 2017) or the World Register of Marine Species (WoRMS; www.marinespecies.org, accessed August 2017). We further excluded species belonging to genera with fossil types denoted by Algaebase, under the condition that these species lacked habitat information on both Algaebase and WoRMS, assuming that the latter species have been collected based on sedimentary or fossilized materials. Species uniquely classified as "freshwater" on both Algaebase and WoRMS were discarded, as these were beyond the scope of our open ocean database. However, we retained species classified as (...)." DR: We add Turland et al. (2018) to the references.

Line 727 ff: "Turland, N. J., Wiersema, J. H., Barrie, F. R., Greuter, W., Hawksworth, D. L., Herendeen, P. S., Knapp, S., Kusber, W.-H., Li, D.-Z., Marhold, K., May, T. W., McNeill, J., Monro, A. M., Prado, J., Price, M. J. & Smith, G. F., editors. International Code of Nomenclature for algae, fungi, and plants (Shenzhen Code) adopted by the Nineteenth International Botanical Congress Shenzhen, China, July 2017. Regnum Vegetabile, Vol. 159. pp. [i]-xxxviii, 1-253. Glashütten: Koeltz Botanical Books, 2018. doi:10.12705/Code.2018."

**DR:** We now also include M. D. Guiry as co-author on the revised manuscript. Line 3: "Damiano Righetti1, Meike Vogt1, Niklaus E. Zimmermann2, Michael D. Guiry3, Nicolas Gruber1" 3AlgaeBase, Ryan Institute, NUI, Galway, University Road, Galway H91 TK33, Ireland

2.3.2. Data merger and synthesis

- Line 270: The rgbif R package should be properly cited and referenced. In addition, please note that there is a typo with the package name ('rgibf' instead of rgbif).

**DR:** Excellent catch. rgbif has now been spellchecked and cited.

Line 275: "(...) using the function datasets in the R package rgbif (Chamberlain, 2015)(...)" Line 609ff: Chamberlain, S.: rgbif: Interface to the Global Biodiversity Information Facility API. R package version 0.9.7. https://cran.r-project.org/package=rgbif, 2015.

3. Results

3.1. Data

- This section is very welcome and acknowledged.

3.1.1. Spatiotemporal coverage

- Line 283: It is recommended to add the EPSG code of the World Geodetic System (WGS84). In addition, I recommend to include this as standard Darwin Core column in PhytoBase using the term geodeticDatum.

**DR:** We now mention the EPSG code in the first instance in the MS: Line 152ff: "However, the latter criterion was fulfilled by all records, as these were standardized to -180 to 180 degrees longitude (rather than 0 to 360 longitude East) and -90 to 90 degrees latitude (WGS84)."

WGS84 had also been included in the Excel sheet (for columns: decimalLatitude, and

decimalLongitude), which accompanies PhytoBase on Pangaea. We consider this information redundant with an additional column added to PhytoBase and prefer to keep the number of columns in the database to the minimum possible, since this increases the usability of the data set, and facilitates treatment of data in analysis software packages.

**5. Data availability**

- In principle, it is highly recommended, based on principles of open and reproducible science and sustainability, that authors make available already the R scripts together with the PhytoBase on PANGAEA, and avoid provision on demand through emails to the authors.

**DR:** We agree with this point. We now provide all 21 R scripts used to do download, clean, and synthesize PhytoBase (and to match data columns with Darwin core terms) through gitlab: https://gitlab.ethz.ch/phytobase/supplementary. Due to the large amount of scripts required to perform each successive step of the database assembly, we gather the scripts into two folders, i.e., "download\_and\_prepare\_data" and "merge\_and\_harmonize\_data".

**Reviewer 3**:**

The MS entitled "PHYTOBASE: A global synthesis of open ocean phytoplankton occurrences" by Righetti et al. represents an interesting effort of combining major existing marine phytoplankton diversity information gathered by microscopy observation, discrimination, identification and, for some of them cells and colony counts, all over ocean systems around the Globe. The authors take into account not only abundance (quantitative) but also presence (qualitative) information in the same database, as well as different sampling methodologies which have an impact on the results obtained, considering bigger or smaller organisms (according to mesh/silk size discrimination and/or microscopy limitations), delicate or robust species (which will not be disrupted by mesh collection), rare or abundant species (depending on the volume of sample analysed). The description of the data as well as the combination methodology, quality control, flagging and taxonomic relevance/correction of the datasets before and after merging them, are clear. The authors make it possible to address a more complete picture by providing a direct and easier access to current knowledge of phytoplankton distribution all over the oceanic realm, identifying properly the uneven distribution od sampling effort and, consequently, of biodiversity assessment or phytoplankton in large areas mainly identified in the Southern Hemisphere. Moreover, they made also an assessment of which are the taxa well known in comparison which the taxa relatively poorly known, mainly concerning small phytoplankton. Finally, they clearly demonstrate the new possibilities in developing ecological models and predictions on the distribution of phytoplankton taxa in open ocean systems.

I therefore recommend this MS to be published in Earth System Science Data after some small technical corrections (see below).

Some general considerations:

One issue to be reminded is that one cannot state for sure, even considering areas which have been well sampled for decades, that some species are not present in a precise area, mostly because, in the corresponding existing databases, studies combining different sampling approaches and, to some extent, also different approaches for considering either morphology, molecular or functional diversity, are scarce.

It remains important then to make this new database as informative as possible, not only concerning the correct nomenclature to be used (and a big effort for make old and new names was also carried out by the present work) but also by considering biases due to different sampling strategies (either nets or tows, Niskin bottles, continuous pumping at a considered depth). One recommendation would be to maintain taxonomic and phylogenetical research as a complement of routine monitoring efforts, providing more accurate consideration of rare species by considering higher sample volumes, concentration by different manners and, the most important, taxonomist expertise which, combined to molecular phylogeny, will certainly make it possible to extract more information from metabarcoding and metagenomic approaches. Moreover, it is also important to consider also new automated approaches which would make it possible to extend the sampling effort on different platforms, addressing most of the time a most limited taxonomical resolution but recalling on functional diversity which, to some extent, would complete taxonomical information included in a marine phytoplankton global database.

**Interpretation of the aspects raised by Reviewer 3 (RE3):**

We thank for the comments raised by RE3. Indeed, we share the view that omission of rare species is a limitation in our work [e.g., Line 350ff: *"However these estimates only represent the fraction of species detectable via light microscopy, and other methods underlying our database, preferentially omitting very rare or small species (Cermeño et al., 2014; Ser-Giacomi et al., 2018; Sogin et al., 2006)*].

**DR:** We have strengthened the point that several diversity dimensions and methodological approaches combined would amplify the benefit of PhytoBase.

Line 135ff: "Additional data processed by the TARA Oceans or Malaspina expedition (Duarte, 2015) may provide valuable context for a future synthesis, and may eventually combine molecular with traditional approaches, yet here we have focused on (...)."

**DR:** We also strengthen the discussion about potential species omission:

Line 483ff: "Second, sampling priorities with respect to taxonomic groups, size classes or species resolution differ widely between research cruises and programs. While small or fragile species may escape detection by the CPR program (Richardson et al., 2006), the resolution of seawater samples is influenced by sampling volume and taxonomic expertise (Cermeño et al., 2014). Our results show that (...)."

Finally, we highlight the benefit of integrating molecular data, in line with the point by RE3: Line 512ff: "The detection of rare species and their integration into PhytoBase may become possible via molecular methods (Bork et al., 2015; Sogin et al., 2006). DNA sequencing has become an alternative approach to (...)."

Some details:

Page 3 line 74: ". . .onto a 270 μm silk roll. . ." as it is important to remind the particular sampling conditions of CPR.

**DR:** We agree and include the detail in mesh size.

Line 74ff: "(...) in which plankton are sampled by filtering seawater onto a silk roll (270  $\mu$ m mesh size) within a recorder device that is towed behind research and commercial ships (Richardson et al., 2006)."

Line 427 ff: "The mesh size of the silk employed in CPR of 270  $\mu$ m under-samples small phytoplankton species (<10  $\mu$ m)."

Page 6 line 170; what about other essential metadata as "collection device" and "analytical tool" (type of microscope) and "volume analysed"? Would this information be available/included/easy to access?

**DR:** In line with the need to retrieve metadata (depending on the purpose of analysis) we retained datasetKeys, resourceIDs and cruiseIDs that link back to specific source archives in PhytoBase as separate columns. Unfortunately, essential metadata on the specific sample collection method are, more often than not, not automatically included in the data retrieved from archives such as GBIF and OBIS. Essentially, we would need to check every dataset key (GBIF) or resourceID (obis), which potentially links metadata with individual datasets in these archives. We consider the inclusion of this information for all taxa considered beyond the scope of this work. Yet, we now refer more explicitly to the option to retrieve metadata: Line 205: "*§§* datasetKey\_gbif and resourceID\_obis are keys to access metadata of original datasets in GBIF and OBIS via API, including information on sampling methods." Line 494ff: "Thus, without careful screening and checking of the data (via e.g. datasetKeys for GBIF records, resourceIDs for OBIS records), the characterization of biogeographies at the species level might be highly biased."

Page 16: Figure 5 caption: "...temperate seas...of Southern Hemisphere (E), cold seas ...of Southern Hemisphere (F)..."

**DR:** The caption has been corrected.

Page 18 lines 419-420: what about other biases of CPR collection as fragile unarmored species, small but also big as ciliates? An extra comment on this issue will be welcomed, as these surveys are one of the most sustained and complete surveys of plankton in some targeted areas.

**DR:** We agree with RE3 that the CPR data contain methodological limitations, with influence the database collected, meaning that fragile or unarmored species, as also rare species, will be underrepresented in the present study. We added additional explanation and discussion with regard to this – and other – sources of bias in our manuscript. Please see our adjustments above, in response to the first (general) comment of RE3.

Page 20 Figure 8 caption: References García et al. 2013; Locarinio et al., 2013 and de Boyer Montegut, 2004 are missing from the reference list. **DR:** The references have been included. Page 22 line 500: To what extent DNA sequencing have really become an alternative to microscopy for characterizing phytoplankton biogeography instead of a complementary and, to some extent supplementary to morphological microscopic identification? **DR:** In our view, this is not a question that can be conclusively addressed. We are in close collaboration with e.g. members of the TARA consortium, and believe that in the future, data collection will tend towards the collection and analysis of environmental (meta)genomic samples, with a move away from traditional microscopy. We believe that classical morphological identification is essential to validate metagenomic information, especially with regard to abundance, biomass or dominance of species. We believe that a merger of traditional and metagenomic data in terms of presence/absence data will be possible, but further efforts need to be made, as come 30% of all oceanic metagenomic data is currently taxonomically unassigned (de Vargas et al., 2015). However, metagenomic data may give us better information eventually on rare and morpholoigically indistinguishable taxa, such as e.g. the vast diversity of picophytoplankton (some of which are included in PhytoBase via MareDat) or haptophytes that cannot be identified using traditional methods.

**DR:** Our view that metagenomic data and traditional data have become *complementary* approaches to characterize phytoplankton biogeography is reflected in the following edit: Line 516ff: "However, we expect that an integration of detailed genetic data with traditional sampling data may soon become possible, allowing to combine several methodological or taxonomic dimensions in databases."

Page 23 line 535: to what extent have you only considered photosynthetical microbial organisms only, especially in some major taxa where both heterotrophs and pigmented cells (mixotrophs or autotrophs) occur? Thanks for precising this in the Materials and Methods section.

**DR:** It is currently not known how much heterotrophy is involved in algae in general, but it is well known that mixotrophy is an issue for the dinoflagellates. We modify the Materials and Methods section to include information with regard to this aspect:

Line 114ff: "This selection of phyla or classes strived to include all autotrophic marine phytoplankton taxa (de Vargas et al., 2015; Falkowski et al., 2004), but it is clear that some of the species may be mixotrophic, particularly for the Dinophyceae (Jeong et al, 2010)."

**PHYTOBASE:** A global synthesis of open ocean phytoplankton occurrences**

Damiano Righetti1, Meike Vogt1, Niklaus E. Zimmermann2, Michael D. Guiry3, Nicolas Gruber1

1Environmental Physics, Institute of Biogeochemistry and Pollutant Dynamics, ETH Zürich, Universitätstrasse 16, 8092
 5Zürich, Switzerland
 2Dynamic Macroecology, Landscape Dynamics, Swiss Federal Research Institute WSL, 8903 Birmensdorf, Switzerland
 3AlgaeBase, Ryan Institute, NUI, Galway, University Road, Galway H91 TK33, Ireland

Correspondence to: Damiano Righetti (damiano.righetti@env.ethz.ch)

Abstract. Marine phytoplankton are responsible for half of the global net primary production and perform multiple other ecological functions and services of the global ocean. These photosynthetic organisms comprise more than 4300 marine species, but their biogeographic patterns and the resulting species diversity are poorly known, mostly owing to severe data limitations. Here, we compile, synthesize, and harmonize marine phytoplankton occurrence records from the two largest biological occurrence archives (Ocean Biogeographic Information System; OBIS, and Global Biodiversity Information Facility; GBIF) and three independent recent data collections. We bring together over 1.36 million phytoplankton occurrence

- 15 records (1.28 million at the level of species) for a total of 1704 species, spanning the principal groups of the diatoms, dinoflagellates, and haptophytes, as well as several other groups. This data compilation increases the amount of marine phytoplankton records available through the single largest contributing archive (OBIS) by 65%. Data span all ocean basins, latitudes and most seasons. Analyzing the oceanic inventory of sampled phytoplankton species richness at the broadest spatial scales possible, using a resampling procedure, we find that richness tends to saturate in the pantropics at ~93% of all
- 20 species in our database, at ~64% in temperate waters, and at ~35% in the cold Northern Hemisphere, while the Southern Hemisphere remains underexplored. We provide metadata on the cruise, research institution, depth and date for each data record, and we include phytoplankton cell counts for 193 763 entries. We strongly recommend consideration of spatiotemporal biases in sampling intensity and varying taxonomic sampling scopes between research cruises or institutions when analyzing the occurrence data spatially. Including such information into statistical analysis tools, such as species
- 25 distribution models may serve to project the diversity, niches, and distribution of species in the contemporary and future ocean, opening the door for quantitative insights into macroecological phytoplankton patterns. PhytoBase can be downloaded from PANGAEA, doi:10.1594/PANGAEA.904397 (Righetti et al., 2019a).

**1** Introduction**

Phytoplankton are photosynthetic members of the plankton, responsible for about half of the global net primary production

30 (Field et al., 1998). While more than 4300 phytoplankton species have been described (Sournia et al., 1991), spanning at least six major clades (Falkowski et al., 2004), there are likely many more species living in the ocean, perhaps more than

10000 (de Vargas et al., 2015). Some of these species (e.g. *Emiliania huxleyi*, *Gephyrocapsa oceanica*) are abundant and occur throughout the ocean (Iglesias-Rodríguez et al., 2002), but a majority of plankton species form low abundance populations (Ser-Giacomi et al., 2018) and remain essentially uncharted; i.e., the quantitative description of where they live,

- 35 and where not, is rather poor. This biogeographic knowledge gap stems from a lack of systematic global surveys, as have been undertaken for inorganic carbon (WOCE/JGOFS/GOSHIP; Wallace 2001) or for trace metals (GEOTRACES; Mawji et al. 2015). Owing to logistic and financial challenges associated with internationally coordinated surveys, our knowledge of phytoplankton biogeography is, with a few exceptions (Bork et al., 2015; McQuatters-Gollop et al., 2015), mostly based on spatially very limited surveys or basin scale studies (e.g., Endo et al., 2018; Honjo and Okada, 1974). Marine phytoplankton
- 40 occurrence data are unevenly distributed, incomplete in remote areas, and orders of magnitude higher in more easily accessed areas, especially near coasts (Buitenhuis et al., 2013). Additional factors that have impeded progress in developing a good biogeographic understanding of the phytoplankton are difficulties in species identification, linked to their microscopic body size. This is well reflected in the current geographic knowledge on phytoplankton species richness from direct observations (e.g. Rodríguez-Ramos et al., 2015), 
[revised manuscript text omitted]

- 105 2013), and records from the global TARA Oceans cruise (Villar et al., 2015), which were not included in GBIF or OBIS at the time of data query (closing window, March 2017). While our selection of additional data was not exhaustive, it strived for the inclusion of quality controlled large-scale phytoplankton datasets. Specifically, MareDat represents a previous global effort in gathering marine plankton data for ecological analyses (e.g., Brun et al., 2015; O'Brien et al., 2016), while Sal et al. (2013) and Villar et al. (2015) are unique in aspects of taxonomic standardization and consistency in methodology.
- 110 We retrieved occurrences at the level "species" or below (e.g., "subspecies", "variety" and "form", as indicated by the taxonRank field in GBIF and OBIS sourced data) for seven phyla: *Cyanobacteria*, *Chlorophyta* (excluding macroalgae), *Cryptophyta*, *Myzozoa*, *Haptophyta*, *Ochrophyta*, and *Euglenozoa*. More specifically, within the *Ochrophyta*, we considered the classes *Bacillariophyceae* (diatoms), *Chrysophyceae*, *Dictyochophyceae*, *Pelagophyceae* and *Raphidophyceae*. Within the *Myzozoa*, we considered the class
- 115 Euglenoidea. This selection of phyla or classes strived to include all autotrophic marine phytoplankton taxa (de Vargas et al., 2015; Falkowski et al., 2004), but it is clear that some of the species may be mixotrophic, particularly for the *Dinophyceae* (Jeong et al., 2010). At the genus level, we additionally retrieved occurrences for *Prochlorococcus* and *Synechococcus* from all sources, as the latter two genera are often highly abundant (Flombaum et al., 2013), but rarely determined to the species level. We also retrieved occurrence records for the functionally relevant genera *Phaeocystis, Richelia, Trichodesmium*, and
- 120 the "picoeukaryote" group from MareDat. For simplicity, we treat genera as "species" in statistics herein. For the selected taxa, occurrence data from GBIF and OBIS were first downloaded in December 2015 and updated in February 2017. Specifically, the initial retrieval of the GBIF data occurred on 7 December 2015 (using the taxonomic

backbone from https://doi.org/10.15468/39omei, accessed on 14 July 2015), and the data were updated on 27 February 2017

(using an updated taxonomic backbone, accessed via http://rs.gbif.org/datasets/backbone, released 27 February 2017). The
 125 data from OBIS were first retrieved on 5 December 2015 using the R package *robis* (Provoost and Bosch, 2015) and the
 OBIS taxonomic backbone, accessed on 4 December 2015 via the R packages *RPostgreSQL* (Conway et al., 2015) and
 *devtools* (Wickham and Chang, 2015). Data were updated for the taxa selected on 6 March 2017 (using the OBIS taxonomic backbone, accessed on 6 March 2017 via the same R packages). The update in 2017 expanded the occurrences retrieved from
 GBIF substantially, with over 20 000 additional phytoplankton records stemming from an Australian CPR program alone

- (AusCPR, https://doi.org/10.1016/j.pocean.2005.09.011, accessed via www.gbif.org on 6 March 2017). We retained any GBIF sourced data that were retrieved in 2015, but deleted from GBIF before March 2017 (such as CPR data, with dataset key 83986ffa-f762-11e1-a439-00145eb45e9a). Occurrence data from the TARA Ocean cruise included the *Bacillariophyceae* and *Dinophyceae* (Villar et al., 2015; their Tables W8 and W9). Occurrence data from MareDat included five phytoplankton papers (Buitenhuis et al., 2012; Leblanc et al., 2012; Luo et al., 2012; O'Brien et al., 2013; Vogt et al., 2012). Additional data gradesed by the TARA Ocean or Melagring and difference and by the TARA.
- 135 2012). Additional data processed by the TARA Oceans or Malaspina expedition (Duarte, 2015) may provide valuable context for a future synthesis, and may eventually combine molecular with traditional approaches, yet here we have focused on publicly available sources until March 2017. The raw sources that underpin GBIF and OBIS, and MareDat, represent decades to centuries of efforts spent in collecting phytoplankton data, including a substantial amount of data from the CPR program (Richardson et al., 2006) and a large fraction of data from the AMT program (cruises 1 to 6) (Sal et al., 2013).

**140 **2.2 Data selection**

We excluded occurrences from waters less than 200 m deep (Amante and Eakins, 2009), from enclosed seas (Baltic Sea, Black Sea or Caspian Sea), and from seas with a surface salinity below 20, using the globally gridded (spatial 1° x 1°) monthly climatological data of Zweng et al. (2013). This salinity-bathymetry threshold served to select data from open oceans, excluding environmentally more complex, and often more fertile, near-shore waters.

**145 2.2.1 Data accessed through GBIF and OBIS**

We included GBIF occurrence records on the basis of "human observation", "observation", "literature", "living specimen", "material sample", "machine observation", "observation", and "unknown", assuming that the latter was based on observation. With respect to OBIS data, we included data records on the basis of "O" and "D", whereby "O" refers to observation and "D" to literature-based records. To filter out raw data of presumably inferior quality, records from OBIS and GBIF were

150 removed: (i) if their year of collection indicated >2017 or <1800 (excluding 110 records; <0.001% of raw data), (ii) if they had no indication on the year or month of collection (excluding 7.2% GBIF raw data and 0.9% OBIS raw data) or (iii) if they had geographic coordinates outside the range -180 to 180 for longitude and/or outside -90 to 90 for latitude. However, the latter criterion was fulfilled by all records, as these were standardized to -180 to 180 degrees longitude (rather than 0 to 360 longitude East) and -90 to 90 degrees latitude (WGS84). Records with negative recording depths (0% of GBIF and 6.6% of OBIS raw data) were flagged and changed to positive, assuming that their original sign was mistaken.

**2.2.2 Data accessed through MAREDAT**

We included occurrence records at the species level for the *Bacillariophyceae* (Leblanc et al., 2012) and *Haptophyta* (O'Brien et al., 2013) and species presence records on *Bacillariophyceae* host cells from Luo et al. (2012). Harmonization of *Haptophyta* species names from MareDat (O'Brien et al., 2013) was guided by a synonymy table provided by O'Brien (*pers.*

160 comm.) (Table A1). Harmonization of Bacillariophyceae species names in MareDat was in progress at the time of first data

access (24 August 2015) and completed (Table A2). In addition, we selected all genus and species level records available for *Trichodesmium, Richelia* (Luo et al., 2012), *Phaeocystis* (Vogt et al., 2012), *Synechococcus* (using the data-field "SynmL") and *Prochlorococcus* (using the data-field "PromL") (Buitenhuis et al., 2012). We included genus level records from the latter taxa, as they represent functionally important phytoplankton groups (Le Quéré, 2005), and as information on the

- 165 presence and abundance of their cells or colonial cells often only existed at genus level (Buitenhuis et al., 2012; Luo et al., 2012; Vogt et al., 2012). Across all sources, data on colonial cells were uniquely provided by MareDat, while additional count data on trichomes for the genus *Trichodesmium* may be accessed from Luo et al. (2012). In addition, we retained records on the "picoeukaryote" group, which were not determined to species or genus level (Buitenhuis et al., 2012). For all taxa, we retained records with reported abundances (i.e., cell counts) larger than zero, while excluding records with zero
- 170 entries or missing data (NA), as our database focuses on presence-only or abundance records. Given that data of the MareDat have been scrutinized previously, we flagged rather than excluded data with reported recording before year 1800 (564 records; values 6, 10 or 11) and unrealistic day entries (58 340 records; values -9 or -1).

**2.2.3 Data accessed through Villar et al. (2015)**

We compiled presence records of species of *Bacillariophyceae* and *Dinophyceae* from the tables W8 and W9 of Villar et al.
(2015). We excluded species names containing "cf" (e.g *Bacteriastrum cf. delicatulum*), as such nomenclature is typically used to refer to closely related species of an observed species. We retained all species (n = 3), which contained "group" in their names (e.g. *Pseudo-nitzschia delicatissima group*). *Tripos lineatus/pentagonus complex* was considered as *Tripos lineatus*. The cleaning of spelling variants of original names from Villar et al. (2015) is presented in Table A3.

**2.2.4 Data accessed through Sal et al. (2013)**

180 We considered occurrence records of the *Bacillariophyceae*, *Dictyochophyceae*, *Dinophyceae*, *Haptophyta* and *Peridinea* and at species level or below, using the species name in the final database. These data included 5891 records from 314 species and 543 samples. The dataset of Sal et al. (2013) represents a highly complementary source of phytoplankton occurrence records, i.e., it had no duplicated records with any of the other sources. This data collection contains in situ samples subjected to consistent methodology performed by the same taxonomist.

**185 2.3 Concatenation of source datasets**

190

Column names or data-fields were adjusted and harmonized to establish compatibility in the dimensions of the different source datasets (Table 1). Columns match Darwin Core standard (https://dwc.tdwg.org) where original data structure could be reconciled with this standard, following GBIF and OBIS that widely rely on Darwin Core. Where critical metadata could not be assigned to Darwin Core, we use additional columns (e.g., columns ending in "gbif" present metadata from GBIF). With regard to sampling depth, GBIF raw data contained the field "depthAccuracy" (18.6% of data with entries) while OBIS raw data contained the fields "depthprecision" (21.64% of data with entries), "minimumDepthInMeters" (Darwin Core term;

**Table 1: Harmonization of column names (data-fields) between data sources and final column name structure in PhytoBase**

|                  | Final column names |                 |                  |                                             |                 |           |                                     |  |
|------------------|--------------------|-----------------|------------------|---------------------------------------------|-----------------|-----------|-------------------------------------|--|
| GBIF (2015)      | GBIF (2017)        | OBIS (2015)     | OBIS (2017)      | MareDat                                     | Villar et al    | Sal et al | (sources merged)                    |  |
| species          | species            | species         | species          | species                                     | species         | species   | scientificName* 1        |  |
| decimalLongitude | longitude          | longitude       | longitude        | Longitude                                   | Longitude       | Lon       | decimalLongitude*                   |  |
| decimalLatitude  | latitude           | latitude        | latitude         | Latitude                                    | Latitude        | Lat       | decimalLatitude*                    |  |
| year             | year               | yearcollected   | year             | Year                                        | Date            | Date      | year*                               |  |
| month            | month              | monthcollected  | month            | Month                                       | Date            | Date      | month*                              |  |
| day              | day                | daycollected    | day              | Day                                         | Date            | Date      | day*                                |  |
| depth            | depth              | depth           | depth            | Depth                                       | Depth           | Depth     | depth                               |  |
| -                | depthAccuracy      | depthprecision  | depthprecision   | -                                           | -               | -         | depthAccuracy                       |  |
| taxonRank        | taxonRank          | -               | -                | rank                                        | -               | -         | taxonRank* ,†            |  |
| -                | occurrencestatus   | -               | occurrencestatus | -                                           | -               | -         | occurrenceStatus*                   |  |
| phylum           | phylum             | phylum          | phylum           | -                                           | -               | -         | phylum* .‡               |  |
| class            | class              | class           | class            | -                                           |                 |           | class *,‡                |  |
| basisOfRecord    | basisOfRecord      | basisofrecord   | basisOfRecord    | -                                           | -               | -         | basisOfRecord*                      |  |
| -                | institutionCode    | institutioncode | institutionCode  | -                                           | -               | -         | institutionCode*.§                  |  |
| -                | -                  | -               | -                | -                                           | -               | -         | sourceArchive                       |  |
| datasetKey       | datasetKey         | -               | -                | -                                           | -               | -         | datasetKey_gbif ll.§§    |  |
| publishingOrgKey | -                  | -               | -                | -                                           | -               | -         | publishingOrgKey_gbif §  |  |
| -                | -                  | collectioncode  | collectionCode   | -                                           | -               | -         | collectionCode_obis II   |  |
| -                | -                  | -               | resname          | -                                           | -               | -         | resname_obis II          |  |
| -                | -                  | resource_id     | resource_id      | -                                           | -               | -         | resourceID_obis II.§§    |  |
| -                | -                  | -               | -                | Origin Database                             | -               | -         | originDatabase_maredat § |  |
| -                | -                  | -               | -                | CruiseorStationID                           | -               | -         | cruiseOrStationID_                  |  |
|                  |                    |                 |                  |                                             |                 |           | maredat ∥                |  |
| -                | -                  | -               | -                | -                                           | Station         | -         | taraStation_villar ll    |  |
| -                | -                  | -               | -                | -                                           | -               | Cruise    | cruise_sal li            |  |
| -                | -                  | -               | -                | -                                           |                 | SampleID  | sampleID_sal                        |  |
| -                | -                  | -               | -                | - Mixed I                                   | Layer Depth (m) | MLD       | MLD_villar_sal                      |  |
| -                | -                  | -               | -                | cellsL -1 ,cellsmL -1 | -               | organism- | organismQuantity* <and></and>       |  |
|                  |                    |                 |                  |                                             |                 | quantity  | organismQuantityType*               |  |
| -                | individualCount    | -               | observedindivi-  | -                                           | -               | -         | individualCount* ,+      |  |
| -                | -                  | -               | dualcount]       | -                                           | -               | -         | yearOfDataAccess                    |  |
| -                | -                  | -               | -                | -                                           | -               | -         | flag                                |  |

GBIF data were downloaded in 2015 (www.gbif.org; retrieved 7 December 2015) and 2017 (retrieved 27 February 2017)

OBIS data were downloaded in 2015 (www.iobis.org; retrieved 5 December 2015) and 2017 (retrieved 6 March 2017)

195 Each occurrence record in PhytoBase is uniquely identifiable by the occurrence ID: scientificName, decimalLongitude, decimalLatitude, year, month, day and depth \*Column names following Darwin Core standard (https://dwc.tdwg.org).

1We retain all original scientificName(s) and synonyms used in individual sources as additional columns with the format "scientificNameOriginal\_<source>"

† The "TaxonRank" field indicates the level of taxonomic resolution (species or genus) of the observation record. Records of subspecies, varieties, and forms were generally extracted from original sources, but considered at the species level (using the genus and specific epithet).

200 #Higher order taxonomy (phylum, class) follows OBIS (taxonomic backbone; retrieved 6 March 2017), which relies on the World Register of Marine Species (www.marinespecies.org).

§ These fields indicate the organization or institution by which original records were collected.

II These fields are indicators of different research cruises or resources, to which original records belonged.

\*"individualCount" and "observedindividualcount" had equivalent entries for records that overlapped between GBIF and OBIS, and were merged into one column.

205 StatasetKey\_gbif and resourceID\_obis are keys to access metadata of original datasets in GBIF and OBIS via API, including information on sampling methods.

25.7% of data with entries) and "maximumDepthInMeters" (Darwin Core term; 24.0% of data with entries). To retain depth precision information from both GBIF and OBIS, we integrated "depthprecision" into the column "depthAccuracy", presented together with a column on "depth" of sampling. To indicate the source from which records were obtained (GBIF, OBIS, MareDat, Villar or Sal) and the year of data access, we added the columns "sourceArchive" and "yearOfDataAccess".

- 210 Last, we added a quality flag column, termed "flag". This column flags records with originally negative collection depth (N) changed to positive (sect. 2.2.1), unrealistic day (D) or year (Y) entries (sect. 2.2.2), and/or records collected from sediment samples or traps (S) rather than seawater samples (sect. 2.3.2). We concatenated the sources into a raw database, which contained 1.51 million depth-referenced occurrence records, 3300 phytoplankton species (including five genera) and 247 385 sampling events (Table 2). Sampling events are thereby (and herein) defined as unique combinations of decimalLongitude,
- 215 decimalLatitude, depth, and time (year, month, day) in the occurrence data.

**2.3.1 Extant species selection and taxonomic harmonization**

We strived for a selection of occurrence data of extant phytoplankton species and a taxonomic harmonization of their multiple spelling variants (merging synonyms, while clearing misspellings or unaccepted names). This procedure included three steps:

- 220 (i) We discarded all species (and their data) that did not have any depth-referenced record. This choice was made on the basis that these species may have been predominantly recorded via fossil materials or have been associated with large uncertainty with respect to their sampling depth, which would infringe the scope of our database.
  - (ii) We extracted all scientific names (mostly at species level, including all synonyms and spelling variants) associated with at least one depth-referenced record from the raw database (Table 2). This resulted in 3300 names, which were validated in August 2017 against the 150 000+ specific and infraspecific names in AlgaeBase (www.algaebase.org), and

matched using a relational database of current names and synonyms; orthography was made as compatible as possible

| Source        | Number of observations
(%unique to source) |         | Number of species*
(%unique to source) |        | Number of observations
(%unique to source) |         | Number of species*
(%unique to source) |        |  |
|---------------|-----------------------------------------------|---------|-------------------------------------------|--------|-----------------------------------------------|---------|-------------------------------------------|--------|--|
|               | full data                                     |         |                                           |        | data with depth-reference                     |         |                                           |        |  |
| GBIF          | 970 927                                       | (65.6)  | 3977                                      | (60.4) | 908 995                                       | (64.2)  | 2676                                      | (51.5) |  |
| OBIS          | 853 981                                       | (60.5)  | 2 305                                     | (25.2) | 823 968                                       | (60.1)  | 1812                                      | (25.4) |  |
| MareDat       | 102621                                        | (94.6)  | 123                                       | (1.1)  | 102467                                        | (94.7)  | 123                                       | (1.5)  |  |
| Villar et al. | 202                                           | (100.0) | 87                                        | (0.0)  | 202                                           | (100.0) | 87                                        | (0.0)  |  |
| Sal et al.    | 5891                                          | (100.0) | 314                                       | (0.0)  | 5867                                          | (100.0) | 313                                       | (0.1)  |  |
| Total         | 1 594 649                                     |         | 4741                                      |        | 1 511 351                                     |         | 3300                                      |        |  |

**Table 2: Summary statistics of the raw database by source**

225

Numbers of observations (with % of observations unique to the source in parentheses) and the numbers of species (with % of species unique to the source in parentheses) presented for each data source. 27 537 observation records of Picoeukaryotes (not identified to species or genus level) are included among the total records and stem from MareDat (all of which contained a depth-reference).
\*Including synonyms or spelling variants.

with the International Code of Nomenclature (Turland et al., 2018), particularly in relation to the gender of specific

- epithets. This screening led to the exclusion of 459 names (and their data), which could not be traced back to any taxonomically accepted name at the time of query, and to the creation of a "synonymy table" in which each original name (including its potentially multiple synonyms and spelling errors) was matched to a corrected or accepted name.
  - (iii) We excluded species (and their data) classified as "fossil only" or "fossil", based on AlgaeBase (www.algaebase.org, accessed August 2017) or the World Register of Marine Species (WoRMS; www.marinespecies.org, accessed August
- 240 2017). We also excluded species belonging to genera with fossil types denoted by AlgaeBase, under the condition that these species lacked habitat information on AlgaeBase, assuming that the latter species have been collected based on sedimentary or fossilized materials. Species uniquely classified as "freshwater" on AlgaeBase were discarded, as these were beyond the scope of our open ocean database. However, we retained species classified as "freshwater", which had at least 24 open ocean (sect 2.2) records and thus were assumed to thrive also in marine habitats: *Aulacoseira granulata, Chaetoceros wighamii, Diatoma rhombica, Dinobryon balticum, Gymnodinium wulffii, Tripos candelabrum, Tripos euarcuatus*. These cleaning steps led to a remaining set of 2032 original species names, synonyms or spelling

variants, corresponding to 1709 taxonomically harmonized species (including five genera not resolved to species level).

**2.3.2 Data merger and synthesis**

- We removed duplicate records, considering the columns "scientificName", "decimalLongitude", "decimalLatitude", "year", 250 "month", "day", and "depth". Removing duplicates meant that any relevant metadata of the duplicated (and hence removed) records were added to the metadata of the record retained, either in an existing or additional column (e.g., information on the original dataset-keys, two which the merged records belonged). We assigned the corrected and/or harmonized taxonomic species name to each original species name in the database on the basis of the synonymy table. We removed duplicates with respect to exact combinations of the harmonized "scientificName", and "decimalLongitude", "decimalLatitude", "year",
- 255 "month", "day", "depth". This resulted in the harmonized database containing 1 360 621 occurrence records (of which 95.8% had a depth-reference), 1709 species (including five genera), and 242 074 sampling events (Table 3). We retained meta-information on the dataset ID, cruise number, and further attributes when removing duplicates. In particular, we retained the original taxonomic name(s) associated with each record in separate columns of type "scientificNameOriginal\_<source>", which allows tracing back the harmonized name to its original name(s). Retaining the original names ensures that future
- 260 taxonomic changes or updated methods for taxonomic harmonization can be readily implemented. Besides the presences, the final database includes 193 777 count records of individuals or cells, spanning 1126 species. Among these, 105 242 records included a volume basis (spanning 335 species), with a predominant origin from MareDat (n = 99498) and Sal et al. (2013) (n = 5744). Last, we flagged sedimentary records, indicated by the column "flag". Although we excluded probably many records based on fossil materials during cleaning step (i), this does not exclude the possibility that occurrence records of
- 265 extant species in the GBIF and OBIS source datasets originated partially from sediment traps or sediment core samples.
  - 9

**Table 3: Summary statistics of the harmonized database by source**

| Source        | Number of ob
(%unique to | servations | Number of species*
(%unique to source) |        | Number of observations
(%unique to source) |                           | Number of species*
(%unique to source) |        |  |  |
|---------------|-----------------------------|------------|-------------------------------------------|--------|-----------------------------------------------|---------------------------|-------------------------------------------|--------|--|--|
|               |                             | full data  |                                           |        |                                               | data with depth-reference |                                           |        |  |  |
| GBIF          | 790 103                     | (54.9)     | 1492                                      | (31.5) | 751 227                                       | (53.7)                    | 1444                                      | (31.3) |  |  |
| OBIS          | 823 836                     | (56.3)     | 1320                                      | (21.6) | 796 907                                       | (56.0)                    | 1283                                      | (22.0) |  |  |
| MareDat       | 101 969                     | (94.7)     | 120                                       | (2.7)  | 101 816                                       | (94.8)                    | 121                                       | (2.7)  |  |  |
| Villar et al. | 202                         | (100.0)    | 87                                        | (0.0)  | 202                                           | (100.0)                   | 87                                        | (0.0)  |  |  |
| Sal et al.    | 5744                        | (100.0)    | 291                                       | (0.0)  | 5721                                          | (100.0)                   | 290                                       | (0.0)  |  |  |
| Total         | 1 360 765                   |            | 1709                                      |        | 1 303 721                                     |                           | 1709                                      |        |  |  |

Numbers of observations (with % of observations unique to the source in parentheses) and numbers of species (with % of species unique to the source in parentheses) presented for each data source.

\*Including 1711 species names and the genera *Phaeocystis, Trichodesmium, Richelia, Prochlorococcus* and *Synechococcus*. 27 537 observation records of Picoeukaryotes (not identified to species or genus level) are included among the total records and stem from MareDat (all of which contained a depth-reference).

Marine sediments can conserve phytoplankton cells that are exported to depth. We flagged phytoplankton records from OBIS and GBIF in the database associated with surface sediment traps or sediment cores (using an "S" in the flag column) by checking the metadata of each individual source dataset of GBIF (using the GBIF datasetKey) and OBIS (using the OBIS

275 resourceID), using the function *datasets* in the R package *rgbif* (Chamberlain, 2015) and the online portal of OBIS (http://iobis.org/explore/#/dataset, accessed 24 October 2018). This check resulted in the flagging of 2.7% of records. We did not attempt to clean or remove sediment type records in MareDat, assuming that information on sampling depth, associated with records of MareDat led to the exclusion of sedimentary records previously. Data from Sal et al. (2013) and Villar et al. (2015) were uniquely based on seawater samples.

**280 3 Results**

**3.1 Data**

**3.1.1 Spatiotemporal coverage**

Phytoplankton occurrence records contained in PhytoBase cover all ocean basins, latitudes, longitudes, and months (Fig. 1). However, data density is globally highly uneven (Fig 1B, C; histograms) with 44.7% of all records falling into the North Atlantic alone, while only 1.4% of records originate from the South Atlantic, and large parts of the South Pacific basin are devoid of records (Fig. 1A). Analyzing the data by latitude (Fig. 1B) and longitude (Fig. 1C) reveals that sampling has been particularly thin at high latitudes (>70°N and S) during wintertime. Occurrences cover a total of 18 863 monthly cells of 1° latitude × 1° longitude (using the World Geodetic System of 1984 as the reference coordinate system; WGS 84), which corresponds to 3.9% of all monthly (n = 12 months) 1° cells of the open ocean (definition; sect. 2.2). Without monthly distinction, records cover 6098 spatial 1° cells, which is a fraction of 15.5% of all 1° cells of the open ocean.